# Contextual Bandits with Entropy-based Human Feedback

## Abstract

In recent years, preference-based human feedback mechanisms have become integral to improving model performance across a range of applications, including conversational AI systems like ChatGPT. However, existing methodologies often overlook critical factors such as model uncertainty and variability in feedback quality. To address these limitations, we propose an innovative entropy-based human feedback framework designed for contextual bandits, which balances exploration and exploitation by soliciting expert feedback when model entropy surpasses a pre-defined threshold. Our method is model-agnostic and adaptable to any contextual bandit agent employing stochastic policies. Through rigorous experimentation, we demonstrate that our approach requires minimal human feedback to achieve significant performance gains, even with suboptimal feedback quality. Our work not only introduces a novel feedback solicitation strategy but also underscores the robustness of integrating human guidance into machine learning systems. Our code is publicly available: `https://anonymous.4open.science/r/CBHF-33C5`

## 1 Introduction

Contextual bandits (CB) have emerged as a powerful framework across various applications, including recommendation systems (Li et al., 2010; Xu et al., 2020), healthcare (Yu et al., 2024), and finance (Zhu et al., 2021), among others (Bouneffouf et al., 2020). CBs enable personalized decision-making by learning from the contextual information in each instance. However, current systems often rely heavily on implicit feedback signals, such as clicks, which are inherently biased and incomplete, limiting their ability to fully capture true user preferences (Qi et al., 2018).

To address these challenges, we explore the incorporation of explicit human feedback in a CB setting. Human feedback has shown promise in reinforcement learning by integrating human guidance into the learning process (Christiano et al., 2017; MacGlashan et al., 2017). Incorporating human feedback enables models to generate more accurate and informative responses, improving performance in applications such as conversational AI like ChatGPT (Ouyang et al., 2022; Achiam et al., 2023), and robotics (Osa et al., 2018).

Human feedback can generally be categorized into action-based feedback from human experts (Osa et al., 2018; Li et al., 2023), and preference-based feedback (Christiano et al., 2017; Saha et al., 2023). This work focuses on the latter. Preference-based feedback, where humans indicate their preference between two options selected by the learner, has gained popularity due to its simplicity. However, existing methods fail to address two critical issues: the varying quality of human feedback and the uncertainty in the model's decisions. These factors often result in inefficient learning and suboptimal performance, especially in high-stakes or complex environments. In this work, we aim to answer the key question: **Can we propose a simple yet effective strategy to incorporate preference-based human feedback in contextual bandits?**

A key challenge in CB problems is balancing exploration and exploitation, which becomes more complex with the addition of human feedback. The algorithm must balance this input to avoid over-reliance while ensuring efficient learning. To address this, we propose a simple criterion for feedback solicitation and introduce two methods for incorporating human feedback into CB, evaluating their performance.

We present two feedback settings. In the action recommendation (AR) method, a human expert provides recommended actions for a given context. In the reward manipulation (RM) method, the expert assigns a reward penalty when the learner selects an action not recommended by the expert. Feedback solicitation is based on model uncertainty, quantified by policy entropy, and human feedback is requested when model entropy exceeds a certain threshold.

These additions underscore the key finding of our study: *even low-quality human feedback, when appropriately solicited, can lead to significant performance improvements*.

Our contributions are threefold. First, we propose a framework to integrate human feedback into CB across different environments and analyze the relative performance of two feedback strategies: *action recommendation* and *reward penalty*. Second, we identify limitations in current approaches and introduce an entropy-based criterion to enhance learning. This criterion not only improves performance but also deepens our understanding of how these methods support learning. Finally, we evaluate the impact of expert feedback quality on CB learner performance, showing how varying levels of human recommendation accuracy affect cumulative rewards. Our findings advocate for the inclusion of our methods in decision-making models and expand the understanding of human feedback integration in reinforcement learning.

## 2 RELATED WORKS

**Contextual bandits** Contextual bandits have diverse applications in recommendation systems (Li et al., 2010; Xu et al., 2020), healthcare (Yu et al., 2024), finance (Zhu et al., 2021), and other fields (Bouneffouf et al., 2020). CBs are a variant of the multi-armed bandit problem where each round is influenced by a specific context, and rewards vary accordingly. This adaptability makes CBs valuable for enhancing various machine learning methods, including supervised learning (Sui & Yu, 2020), unsupervised learning (Sublime & Lefebvre, 2018), active learning (Bouneffouf et al., 2014), and reinforcement learning (Intayoad et al., 2020).

To tackle CB challenges, several algorithms have been developed, such as LINUCB (Li et al., 2010), Neural Bandit (Allesiardo et al., 2014), and Thompson sampling (Agrawal & Goyal, 2013). These typically assume a linear dependency between the expected reward and its context. Despite these advancements, CBs often rely on implicit feedback, like user clicks, leading to biased and incomplete evaluations of user preferences (Qi et al., 2018). This reliance complicates accurately gauging user responses and tailoring the learning process.

**Human feedback in the loop** Recent advancements in human-in-the-loop methodologies have shown significant successes in real-life applications, such as ChatGPT via reinforcement learning with human feedback (RLHF) (MacGlashan et al., 2017), as well as in robotics (Argall et al., 2009) and health informatics (Holzinger, 2016).

Preference-based feedback can be categorized into three groups: i) action-based preferences (Fürnkranz et al., 2012), where experts rank actions, ii) state preferences (Wirth & Fürnkranz, 2014), and iii) trajectory preferences Busa-Fekete et al. (2014); Novoseller et al. (2020). Action-based feedback from humans is explored in (Mandel et al., 2017), where experts add actions to a reinforcement learning agent to boost performance. Other forms of explicit human feedback include reward shaping (Xiao et al., 2020; Bıyık et al., 2022; Ibarz et al., 2018; Arakawa et al., 2018). These approaches however do not account for acquiring feedback based on the learner's uncertainty or the impact of varying levels of feedback on performance.

**Contextual bandits with human feedback** Human-in-the-Loop Reinforcement Learning addresses the bias problem of implicit feedback in contextual bandits. The exploration of learning in multi-armed bandits with human feedback is discussed in (Tang & Ho, 2019), where a human expert provides biased reports based on observed rewards. The learner's goal is to select arms sequentially using this biased feedback to maximize rewards, without direct access to the actual rewards.

Preference-based feedback in contextual and dueling bandit frameworks has been explored in previous studies (Sekhari et al., 2023; Dudík et al., 2015; Saha, 2021; Wu et al., 2023). The learner presents candidate actions and receives noisy preferences from a human expert, focusing on minimizing regret and active queries. In contrast, we consider a setup where the learner receives direct feedback from human experts and show how the fraction of active queries varies with different sets of experts.

**Active learning in contextual bandits** Active learning (Judah et al., 2014) enhances performance by selectively querying the most informative data points for labeling, rather than passively receiving labels for randomly or sequentially presented data. In the context of bandit algorithms, active learning has been employed to optimize the exploration-exploitation trade-off by guiding the algorithm to request feedback or labels when it is most uncertain about an action's outcome (Taylor & Stone, 2009). For example, Bouneffouf et al. (2014) integrated active learning with Thompson sampling and UCB algorithms in contextual bandits, resulting in improved sample efficiency.

In our work, we build on this idea by combining active learning techniques with human feedback, utilizing an entropy-based mechanism to query feedback when necessary. By incorporating active learning principles into our contextual bandit framework, we aim to more effectively balance exploration and exploitation, particularly in scenarios where human feedback is noisy or costly. This approach not only improves sample efficiency but also helps mitigate the challenges posed by varying feedback quality.

**Other related areas** Our work builds on several important research areas, including counterfactual reasoning, imitation learning, preference optimization, and entropy-based active learning. We draw inspiration from Tang and Wiens Tang & Wiens (2023), whose counterfactual-augmented importance sampling informs our feedback framework, and extend DAGGER Ross et al. (2011) by dynamically incorporating expert feedback instead of using fixed imitation. We also acknowledge parallels with Active Preference Optimization (APO) Das et al. (2024), adapting trajectory-level preference feedback to reward manipulation in more complex settings. Additionally, we connect with entropy-driven methods like BALD Houlsby et al. (2011) and IDS Russo & Van Roy (2014), adapting their principles for contextual bandit problems to balance information gain and decision-making efficiency in sequential exploration. These connections highlight how our approach advances real-time feedback integration and decision optimization.

## 3 METHOD

The following section provides a description of our method and its subcomponents. A comprehensive representation of the approach is shown in Figure 1. Algorithm 1 describes our method.

### 3.1 CONTEXTUAL BANDIT FORMULATION

We consider an online stochastic contextual bandit framework where at each round $t$, the world generates a context-reward pair $(s_t, r_t)$ sampled independently from a stationary unknown distribution $\mathcal{D}$. Here $s_t \in \mathcal{S} = \mathbb{R}^m$ is an $m$ dimensional real valued vector and $r_t = (r_t(1), \ldots, r_t(k)) \in \{0, 1\}^k$ is a $k$-dimensional vector where each element can take values 0 or 1. The agent then chooses an action $a_t \in \{1, \ldots, k\}$ according to a policy $\pi : \mathcal{S} \mapsto \{1, \ldots, k\}$ and the environment reveals the reward $r_t(a_t) \in \{0, 1\}$.

The objective of the agent is to find a policy $\pi \in \Pi$ that maximizes the expected cumulative reward given by

$$\max_{a_t \sim \pi} \sum_{t=1}^{T} \mathbb{E}\big[r_t(a_t) \mid s_t, a_t\big] \tag{1}$$

The problem setup described above bears a strong resemblance to a multi-label or multiclass classification problem, where $r_t(a_t) = 1$ indicates the correct label choice and 0 otherwise. However, a key distinction lies in the learner's lack of access to the correct label or label set for each observation. Instead, the learner only discerns whether the chosen label for an observation is correct or incorrect.

### 3.2 INCORPORATING ENTROPY BASED HUMAN FEEDBACK

In contextual bandits, feedbacks are provided in the form of a reward signal predetermined by the designer. These reward signals are not well defined for complex decision making problems (Blanchard et al., 2023; Dragone et al., 2019), and are often learned from data. An alternate to learning a reward function from data is to obtain preference based feedback from humans and learn the underlying reward function that the human expert is optimizing (Sekhari et al., 2024). In this work, we consider the setup where human expert has sufficient expertise and valuable insights stemming from their

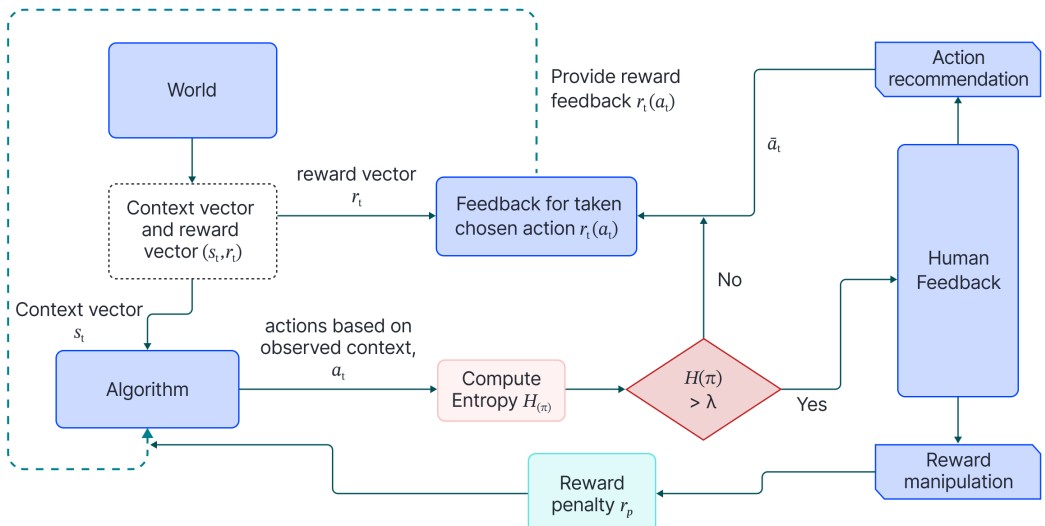

Figure 1: Overview of the proposed architecture.

experience and domain knowledge to provide direct feedback to the learner. These feedbacks can directly impact the actions a contextual bandit learner takes or the rewards it receives. However, the quality of such explicit feedback may vary depending on the expertise levels of different individuals. We provide two ways in which human experts can provide feedback to the contextual bandit learner: i) Action Recommendation through direct supervision (AR) ii) Reward Manipulation (RM). In certain applications, a human expert can directly control the actions that the agent takes; in these cases, feedback in the form of action recommendations (AR) is useful. Conversely, in other applications where the human expert cannot directly influence the agent's actions, feedback through reward manipulation is more beneficial.

We describe each of these different feedback below.

### 3.2.1 ACTION RECOMMENDATION VIA DIRECT SUPERVISION

In this form of feedback, the human expert explicitly instructs the actions to take for a given context. We assume that the algorithm always accepts the recommended action. Let $\hat{a}_t$ be a set of actions recommended by the human expert $\mathcal{E}^{\mathrm{AR}}$ for a given context $s_t$ and expert quality $q_t$, where $q_t \in [0, 1]$, we elaborate more on the expert quality in Section 3.4. When the expert recommends a set of actions, the learning algorithm randomly chooses an action from the recommended set. The final reward $r_t^f$ received by the learner is given by:

$$\hat{a}_t = \mathcal{E}^{\mathrm{AR}}(s_t, q_t) \tag{2}$$
$$a_t \sim \mathrm{Uniform}(\hat{a}_t) \tag{3}$$
$$r_t^f = r_t(a_t) \tag{4}$$

### 3.2.2 REWARD MANIPULATION

In this form of feedback, the human expert $\mathcal{E}^{\mathrm{RM}}$ gives an additional reward penalty when the learner chooses an action not recommended by the expert. Let $r_p$ be the fixed reward penalty for non-recommended actions. Let $a_t$ be the action chosen by the learner at round $t$, and $\hat{a}_t$ be the expert's recommended action set. The final reward $r_t^f$ received by the learner is given by:

---

**Algorithm 1** Enropy Based - CBHF

---

**Require:** Input parameters: entropy threshold $(\lambda)$, feedback-type $(fb)$, round-number $(n)$, contextual bandit agent $(\mathcal{A})$, human expert quality $(q_t)$
**Ensure:** Output: *mean cumulative reward*
 1: Initialize *mean cumulative reward* $\leftarrow 0$
 2: **for** $t = 1$ to $n$ **do**
 3:    Get context, reward vector $(s_t, r_t) \leftarrow \omega$
 4:    Get actions and action distribution from the learner $(a_t, \pi(s_t)) \leftarrow \mathcal{A}(s_t)$
 5:    Compute $H(\pi(s_t))$
 6:    **if** $H(\pi(s_t)) > \lambda$ **then**
 7:       **if** $fb == AR$ **then**
 8:          $\hat{a} \leftarrow \mathcal{E}(s_t, q_t)$
 9:          $a_t \leftarrow \hat{a}$
10:          $r \leftarrow r_t(a_t)$
11:       **else if** $fb == RM$ **then**
12:          $r_p \leftarrow \mathcal{E}(s_t, q_t)$
13:          $r \leftarrow r_t(a_t) + r_p$
14:       **end if**
15:    **else**
16:       $r \leftarrow r_t(a_t)$
17:    **end if**
18:    Update Agent $\mathcal{A}$ policy $\pi$ with feedback $r$
19:    *mean cumulative reward* $\leftarrow$ evaluate agent $\mathcal{A}$
20: **end for**
21: **return** *mean cumulative reward*

---

$$r_p = \mathcal{E}^{\mathrm{RM}}(s_t, q_t) \tag{5}$$

$$r_t^f = \begin{cases} r_t(a_t) + r_p & \text{if } a_t \notin \hat{a}_t \\ r_t(a_t) & \text{otherwise} \end{cases} \tag{6}$$

## 3.3 WHEN TO SEEK HUMAN FEEDBACK?

An important question that naturally arises when integrating human feedback into the contextual bandit algorithm is when the algorithm will actively seek out such feedback. In the contextual duelling bandit setup in (Di et al., 2024), the algorithm presents two options to the human and asks them to choose a preferred one based on a given context. In the case of model misspecification, where the underlying reward function assumed by the algorithm does match the true rewards generated by human preferences, the algorithm can actively query the human expert to obtain feedback on the predicted reward or rankings (Yang et al., 2023). In our work, we take a different approach where the learner seeks for expert feedback based on model uncertainty. The model computes the entropy of the policy at each round $t$ which quantifies the degree of unpredictability in the policy's decision making process using the following expression

$$H(\pi) = -\sum_{a_t} \pi(a_t \mid s_t) \log(\pi(a_t \mid s_t)), \tag{7}$$

where $H(\pi)$ denotes the entropy of policy $\pi$. The model then queries for human feedback when the model entropy exceeds a predefined threshold $\lambda$. Appropriate choice of $\lambda$ will depend on the problem domain and are obtained using hyper parameter search. Our proposed entropy based approach for querying the expert depends on the learner's ability to compute an entropy for its policy. Thus for certain models when model uncertainty is not available, we can still obtain two forms of human feedback periodically, we also demonstrate the effect on model performance when these two types of human feedback are incorporated for different periods.

### 3.4 QUALITY OF EXPERTS

We consider the effect of learner's performance based on different quality of expert feedback received. We define the quality of feedback in this case as the accuracy of the expert in providing correct recommendation. We first show how the performance of the contextual bandit learner measured by the expected cumulative reward varies for different expert levels of accuracy. Let $q_t \in [0, 1]$ be the probability of providing correct recommendation associated with a particular level of expert. During training, the algorithm seeks expert feedback described in Section 3.2.1 and 3.2.2 when $H(\pi) \geq \lambda$. For action recommendation via direct supervision, the expert provides the correct action with probability $q_t$ and provides a randomized action with probability $1 - q_t$. For reward manipulation feedback, the expert wrongly penalizes the learner with a probability of $1 - q_t$.

## 4 EXPERIMENTS

### 4.1 EXPERIMENTAL SETUP

In this sub-section, we present the environment settings, baselines, and experimental results. We also discuss the effect of entropy thresholds and expert accuracy on model performance.

**Algorithms and Environments Considered.** We conduct experiments across a range of *environments* and *contextual bandit agents*. The agents fall into two categories: (i) classic contextual bandit algorithms and (ii) policy-based reinforcement learning (RL) algorithms with a discount factor of 0, focusing on immediate rewards.

**Classic Contextual Bandit Algorithms.** For the classic contextual bandit setup, we employ three key algorithms: 1. **LinearUCB** (Li et al., 2010): An extension of the traditional Upper Confidence Bound (UCB) algorithm (Auer, 2002), where the expected reward for each action depends linearly on the context or features associated with that action. 2. **Bootstrapped Thompson Sampling** (Kaptein & Eckles, 2014): This method replaces the posterior distribution in standard Thompson Sampling with a bootstrapped distribution, enhancing robustness by resampling historical data instead of relying on a parametric model. 3. **EE-NET** (Ban et al., 2021): This approach utilizes two neural networks—one for exploration and one for exploitation—to learn a reward function and adaptively balance exploration with exploitation.

**Policy-Based Reinforcement Learning Algorithms.** For policy-based RL, we evaluate four algorithms, with the discount factor set to 0 to prioritize immediate rewards: **Proximal Policy Optimization (PPO)** (Schulman et al., 2017), **PPO with Long Short-Term Memory (PPO-LSTM)**, **REINFORCE** (Williams, 1992), **Actor-Critic** (Haarnoja et al., 2018).

**Baseline Comparison.** We include the **TAMER framework** (Knox & Stone, 2009) as a baseline, which allows human trainers to provide real-time feedback to the agent, supplementing the predefined environmental reward signal. In our experiments, we simulate human feedback by revealing the true labels during training.

**Expert Feedback Comparison.** For all contextual bandit agents, we compare two types of expert feedback as described in sections 3.2.1 and 3.2.2. Expert feedback is solicited only during the training phase, and each learner is evaluated after five independent runs, with the mean cumulative reward reported.

**Datasets.** We use multi-label datasets from the Extreme Classification Repository, including Bibtex, Media Mill, and Delicious (Bhatia et al., 2016). In the contextual bandit framework, the reward function for these supervised learning datasets is defined as:

$$r_t(a_t) = \begin{cases} 1 & \text{if } a_t \in y_t \\ 0 & \text{otherwise} \end{cases} \tag{8}$$

where $y_t$ represents the set of correct labels associated with context $s_t$. These datasets are selected for their size, complexity, and diversity, making them suitable for evaluating contextual bandits with human feedback.

**Implementation Details.** We consider a range of entropy thresholds as hyperparameters, controlling how frequently the algorithm seeks to incorporate human feedback. The specific ranges for different

datasets are detailed in Appendix E.2. We select the optimal entropy threshold and report the mean cumulative reward for each mode of human expert feedback. The code base for policy-based RL algorithms is implemented in PyTorch, adapted from (seungeunrho, 2019), while the LinearUCB and Bootstrapped Thompson Sampling implementations are adapted from (Cortes, 2019). The hyperparameters for the RL algorithms are provided in Appendix E.1. Additionally, expert quality is varied based on values of $q_t \in [0, 1]$, where with probability $q_t$, the correct label or set of labels associated with context $s_t$ is provided to the learner, as mentioned in Section 3.3.

## 4.2 VARIATION OF MODEL PERFORMANCE BASED ON DIFFERENT EXPERT QUALITY

We first present the effect of different expert quality on the two types of feedback discussed in Section 3.2.1 and Section 3.2.2. Note that we can compute the entropy of policy $\pi$ for the PPO, PPO-LSTM, Reinforce, Actor-Critic and LinearUCB and Bootstrapped Thompson sampling. We now present the results associated with different expert levels in for the four environments discussed in section 4. Figure 2 shows the variation of different expert qualities for different range of learners. The bar plot in orange shows the model performance when reward manipulation is used as a feedback from the human expert and the bar plot in blue shows the model performance when action recommendation as a feedback from human feedback. Our analysis shows that for different expert levels the effectiveness of incorporating human feedback depends on the learner. Comparison of expert levels with model performance for other learners are shown in Appendix A.

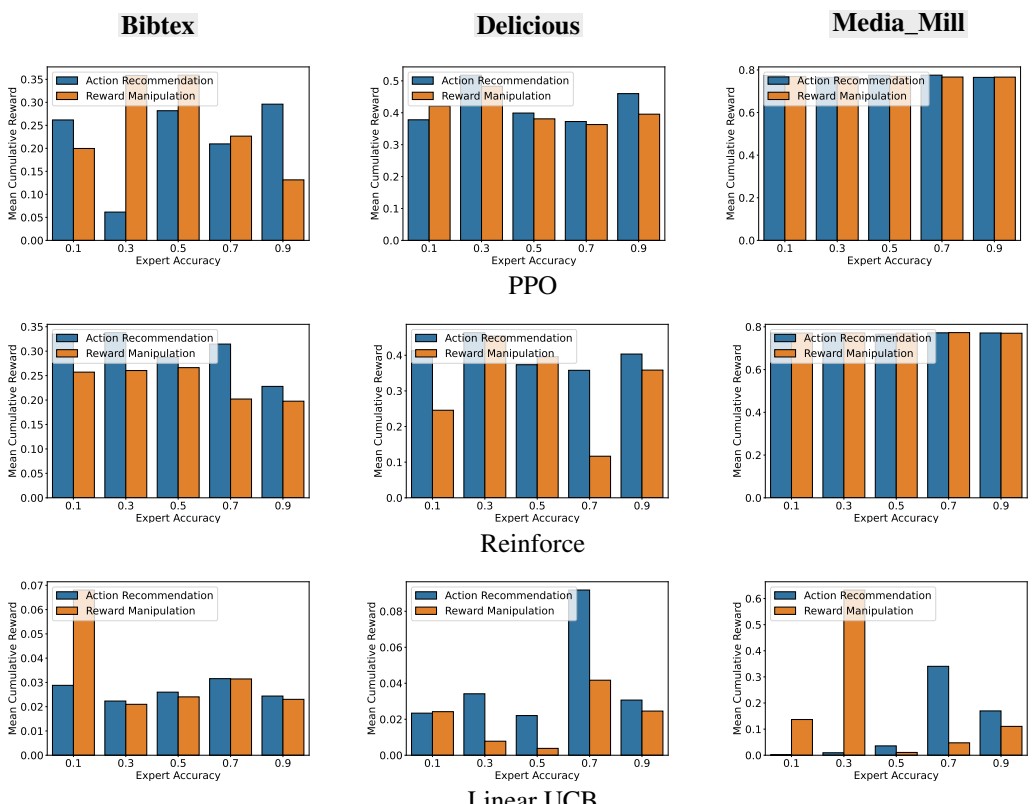

Figure 2: Comparison of expert feedback for different learners based on different expert qualities. The results show that mean cumulative reward for different datasets and algorithms vary in a different manner for the two feedback schemes considered. Higher levels of expert does not necessary results in better performance.

Table 1: Performance comparison of algorithms for different quality of expert feedback. The values in bold represent the maximum mean cumulative reward achieved across different levels of expert.

| Feedback Type | Algorithm Name | Environment Name | 0.3 | 0.5 | 0.7 | 0.9 |
|---|---|---|---|---|---|---|
| Action Recommendation | PPO | Bibtex | **0.27349 ± 0.00167** | 0.26383 ± 0.00091 | 0.20268 ± 0.00260 | 0.16763 ± 0.00092 |
| Reward Manipulation | PPO | Bibtex | 0.27827 ± 0.00312 | 0.27470 ± 0.00165 | 0.16965 ± 0.00202 | **0.31021 ± 0.00278** |
| Action Recommendation | PPO | Delicious | **0.51770 ± 0.00220** | 0.36824 ± 0.00191 | 0.37114 ± 0.00208 | 0.46170 ± 0.00130 |
| Reward Manipulation | PPO | Delicious | **0.48187 ± 0.00113** | 0.29682 ± 0.00230 | 0.36717 ± 0.00215 | 0.40190 ± 0.00165 |
| Action Recommendation | PPO-LSTM | Media_Mill | 0.76836 ± 0.00155 | 0.77318 ± 0.00141 | **0.77504 ± 0.00058** | 0.77113 ± 0.00120 |
| Reward Manipulation | PPO-LSTM | Media_Mill | 0.76973 ± 0.00114 | **0.77447 ± 0.00177** | 0.76748 ± 0.00187 | 0.76197 ± 0.00373 |
| Action Recommendation | LinearUCB | Bibtex | **0.02478 ± 0.00068** | 0.02280 ± 0.00056 | 0.02145 ± 0.00066 | 0.02002 ± 0.00055 |
| Reward Manipulation | LinearUCB | Bibtex | 0.02369 ± 0.00080 | 0.02532 ± 0.00079 | 0.02518 ± 0.00049 | **0.03527 ± 0.00115** |
| Action Recommendation | LinearUCB | Delicious | **0.02430 ± 0.00053** | 0.01818 ± 0.00036 | 0.02064 ± 0.00061 | 0.05308 ± 0.00066 |
| Reward Manipulation | LinearUCB | Delicious | 0.01664 ± 0.00022 | **0.10018 ± 0.00161** | 0.01889 ± 0.00051 | 0.08540 ± 0.00063 |
| Action Recommendation | Bootstrapped-TS | Bibtex | 0.22537 ± 0.00196 | 0.19911 ± 0.00105 | 0.21668 ± 0.00144 | **0.24097 ± 0.00137** |
| Reward Manipulation | Bootstrapped-TS | Bibtex | 0.15276 ± 0.00101 | **0.27697 ± 0.00103** | 0.18423 ± 0.00087 | 0.18468 ± 0.00278 |

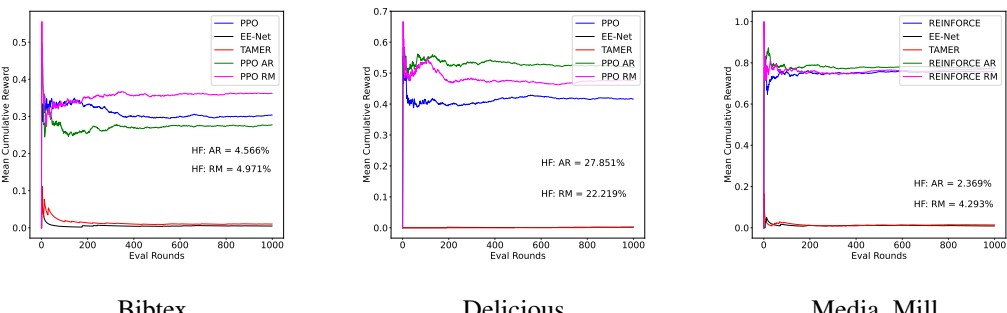

| Bibtex | Delicious | Media_Mill |

Figure 3: Performance comparison with baselines. Human feedback consistently leads to large performance gains.

### 4.3 INCORPORATING ENTROPY BASED FEEDBACK ACHIEVES HIGHER PERFORMANCE COMPARED TO BASELINES

We optimize the model performance across various expert levels and compare these results with baseline models, including TAMER and EE-Net. Figure 3 presents the mean cumulative reward for the optimized expert level (as obtained from Table 1), highlighting the significant performance gains achieved by incorporating entropy-based feedback over the baselines.

Our analysis, conducted across all datasets, demonstrates that integrating entropy-based feedback—specifically Action Recommendation (AR) and Reward Modification (RM)—consistently outperforms both TAMER and EE-Net. Moreover, we observe that the proportion of steps during which the algorithm seeks human expert feedback varies across datasets. Importantly, the results reveal two key findings:

Firstly, learners benefit substantially from entropy-based feedback compared to when no such feedback is provided. This improvement underscores the effectiveness of entropy thresholds in selectively involving human experts, thereby guiding the learning process. In fact, even with a modest number of queries to the human expert (less than 30% of the total training steps), entropy-based feedback drives superior performance over the baseline models. Secondly, the final performance of the learners is not strictly dependent on the quality of the human feedback, as shown in Figure 2.

Interestingly, the performance of AR and RM varies between datasets. For example, on the Bibtex dataset, AR performs worse compared to RM, while on the Delicious dataset, AR demonstrates the best performance among the three. This difference arises due to how penalties affect exploration: Bibtex, with fewer actions, benefits less from AR's action-space limitation, whereas Delicious, with many possible actions, sees AR accelerating convergence by narrowing down the action space early in the learning process. As a result, AR's advantage becomes more apparent in environments where an overwhelming number of actions could otherwise slow down the learner's progress.

Further details regarding the proportion of expert queries for different levels of expert quality are provided in Appendix C.

## 4.4 EFFECT OF ENTROPY THRESHOLD AND EXPERT ACCURACY ON MODEL PERFORMANCE

Figure 4 presents bubble plots comparing model performance at different expert levels and entropy threshold values for both AR and RM feedback types. The size and color of each bubble represent the mean cumulative reward for the corresponding learner.

We begin by analyzing the results for AR feedback. Generally, we observe that at higher entropy threshold values, the model's performance remains relatively stable across different expert levels. This behavior is expected, as higher entropy thresholds result in fewer queries to the human expert, reducing the impact of expert quality on performance.

However, at lower entropy thresholds, an interesting pattern emerges: increasing expert quality can actually lead to a decrease in model performance. This phenomenon relates to the exploration-exploitation trade-off. At high expert levels, the expert consistently provides accurate recommendations, and since the model is designed to always accept these recommendations in the AR setting, the result is pure exploitation. Conversely, at lower expert levels, where recommendations are more random, the model is encouraged to explore a broader set of actions, which can ultimately yield higher cumulative rewards.

A similar pattern is observed with RM feedback. At higher entropy thresholds, the differences in performance between varying expert levels are minimal, as fewer queries are made to the expert. At lower entropy thresholds, however, we again see a decline in performance as expert quality increases.

Further bubble plots illustrating these trends for other learners, under both AR and RM feedback, can be found in Appendix B.

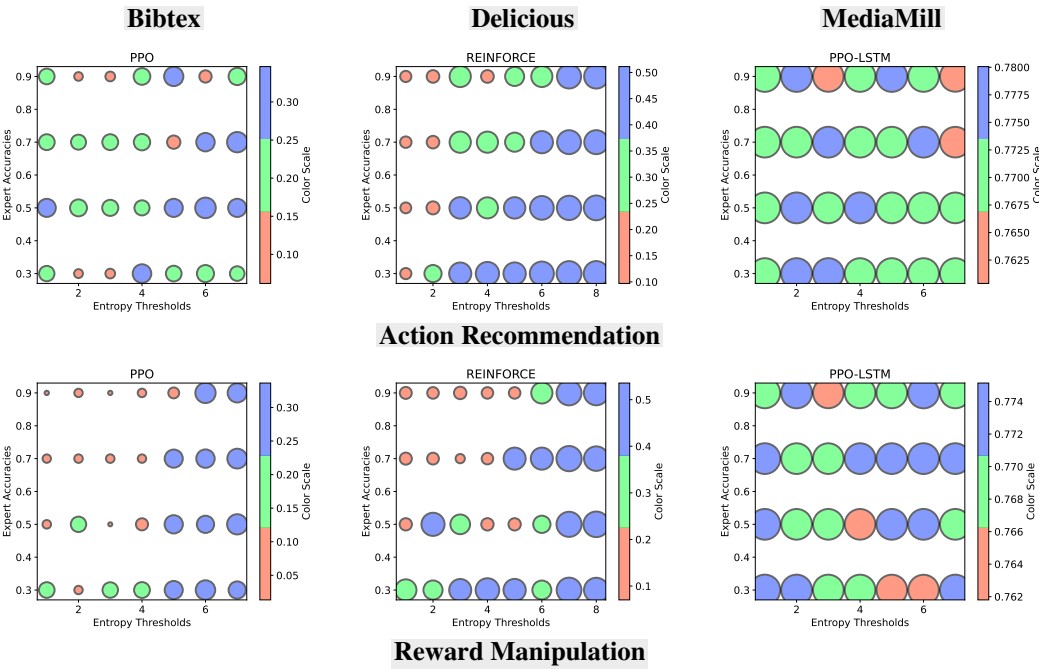

Figure 4: Comparison of model performance for different values of entropy and expert accuracies for feedback: Action Recommendation and Reward Manipulation. The size and color of each bubble in the bubble plots represent the magnitude of the mean cumulative reward.

## 4.5 OBSERVED DIFFERENCES BETWEEN FEEDBACK TYPES

Figure 3 illustrates how the two forms of feedback, AR and RM, interact differently with the underlying algorithms and datasets. The choice of feedback type should therefore depend on the specific application.

Our results generally indicate that at higher expert levels, AR tends to be more effective than RM. This is likely because AR directly influences the actions taken by the contextual bandit (CB), interfering less with its reward-based learning process. At low expert levels, however, AR can become disruptive, leading to poor exploration by prematurely narrowing the action space. In contrast, at high expert levels, AR provides clearer guidance for the bandit's exploration, optimizing action selection while leaving the reward structure relatively intact.

Ultimately, this suggests that AR is particularly advantageous when expert quality is high, as it can effectively guide exploration without destabilizing the learning process.

## 5 CONCLUSION

In conclusion, this work introduces an effective entropy-based framework for incorporating human feedback into contextual bandits. By utilizing model entropy to trigger feedback solicitation, we significantly reduce the reliance on continuous human intervention, thus making the system more efficient and scalable. Our experiments show that even with low-quality human feedback, substantial performance gains can be achieved, underscoring the potential of entropy-based feedback mechanisms in various real-world applications. This framework enhances learning efficiency and provides new insights into the dynamics of human-machine collaboration in reinforcement learning environments. Future work may focus on refining feedback solicitation strategies and exploring their applicability in broader AI contexts, ensuring even more adaptive and responsive learning systems.

## 6 IMPACT STATEMENT

This paper presents work whose goal is to advance the field of Machine Learning. There are many potential societal consequences of our work, none which we feel must be specifically highlighted here.

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

## A EFFECT OF FEEDBACK QUALITIES ON DIFFERENT LEARNERS

This section details the impact of feedback levels on different learners.

Figure 5: Comparison of expert feedback for different learners based on different expert qualities. The results show that mean cumulative reward for different datasets and algorithms vary in a different manner for the two feedback schemes considered. Higher levels of expert does not necessary results in better performance.

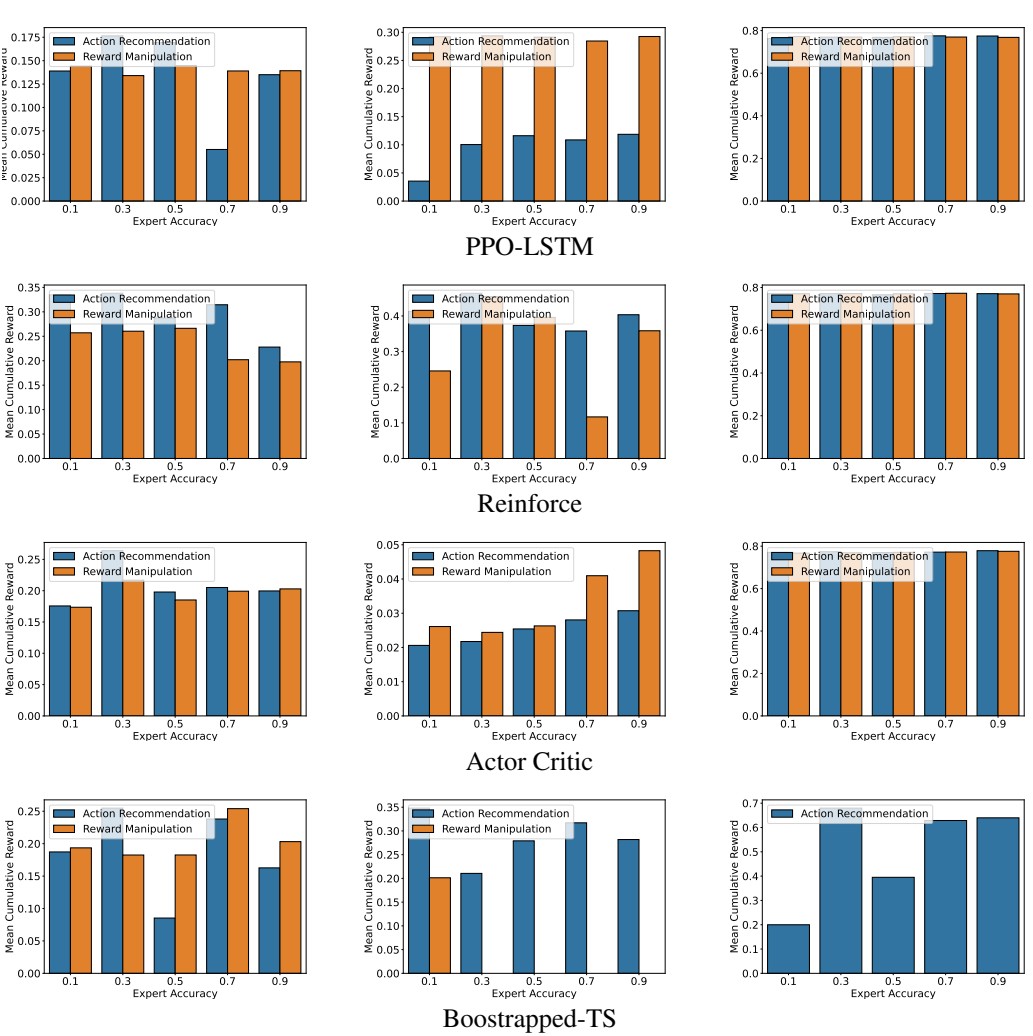

## B EFFECT OF ENTROPY THRESHOLD AND EXPERT LEVELS ON MODEL PERFORMANCE

This section studies the impact of the entropy threshold on performance.

Figure 6: Comparison of model performance for different values of entropy and expert accuracies for feedback: Action Recommendation. The size and color of each bubble in the bubble plots represent the magnitude of the mean cumulative reward.

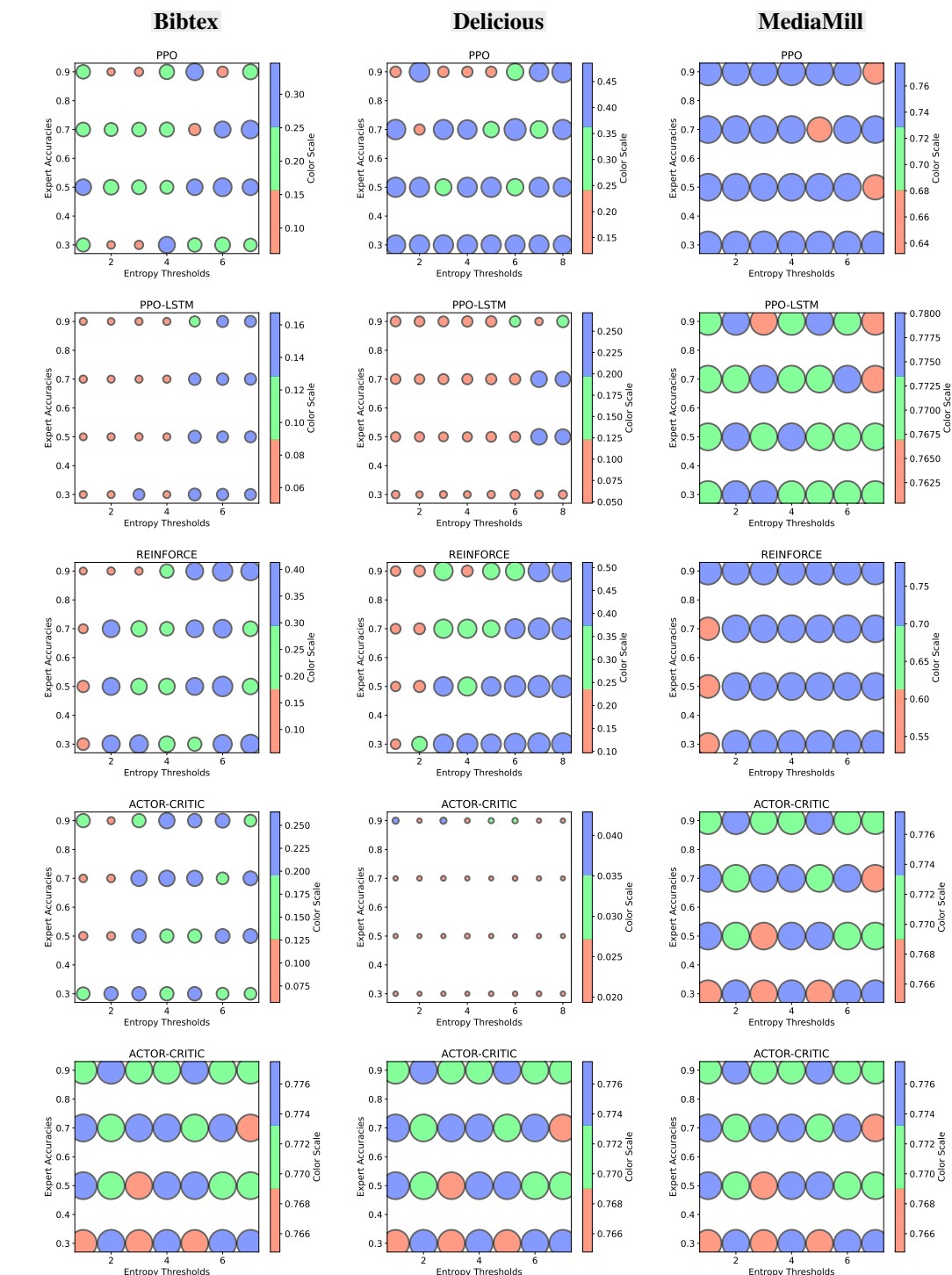

Figure 7: Comparison of model performance for different values of entropy and expert accuracies for feedback: Reward Manipulation. The size and color of each bubble in the bubble plots represent the magnitude of the mean cumulative reward.

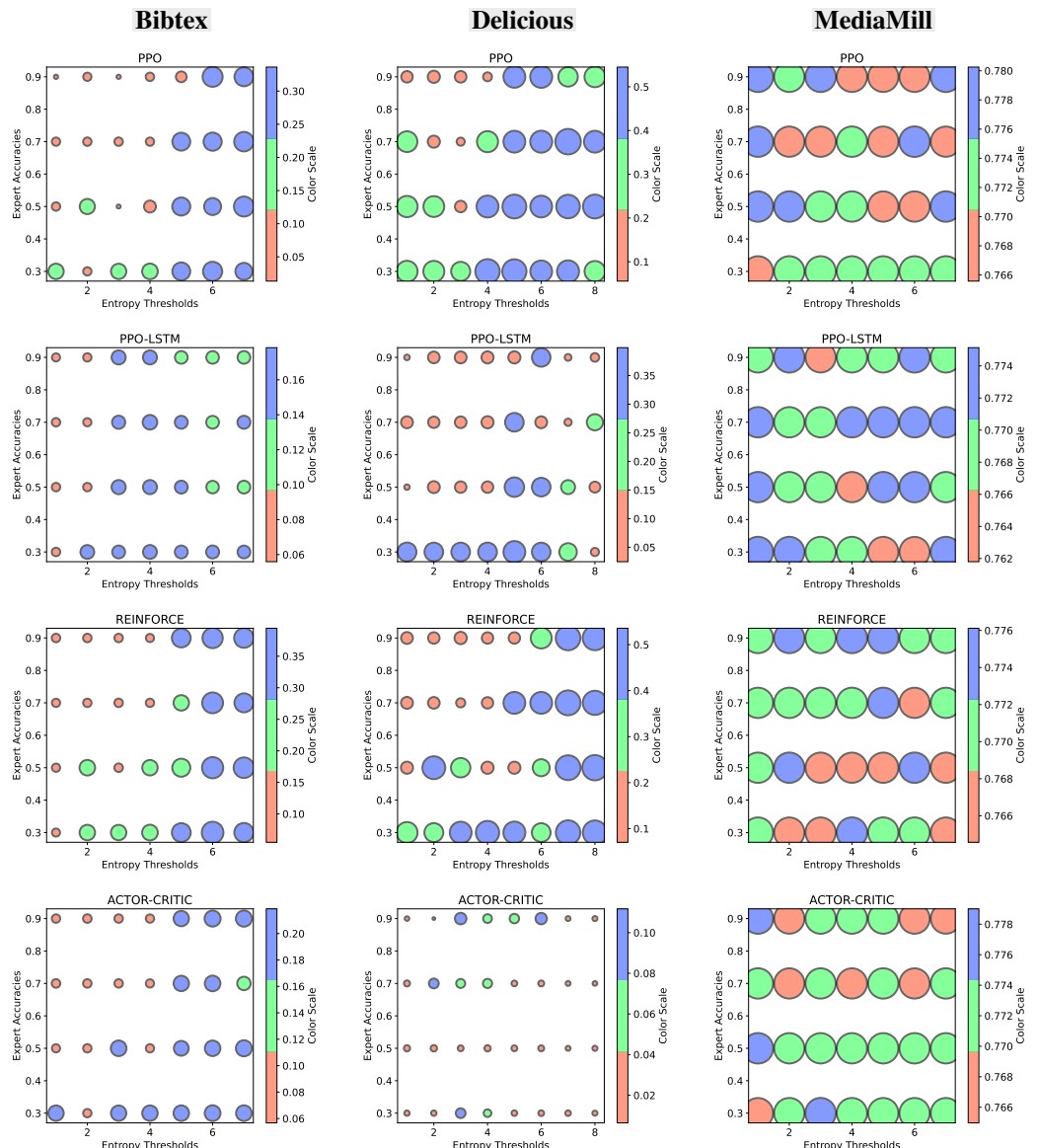

## C    VARIATION IN THE PERCENTAGE OF STEPS FOR EXPERT QUERIES BASED ON ENTROPY THRESHOLD

This section studies the variation of expert queries for the two feedback types.

## D    PERFORMANCE OF DIFFERENT ALGORITHMS BASED ON DIFFERENT EXPERT LEVELS

This section provides more details on the performance as a function of different expert levels.

Figure 8: Variation of expert queries made for different models based on entropy for feedback type: Action Recommendation

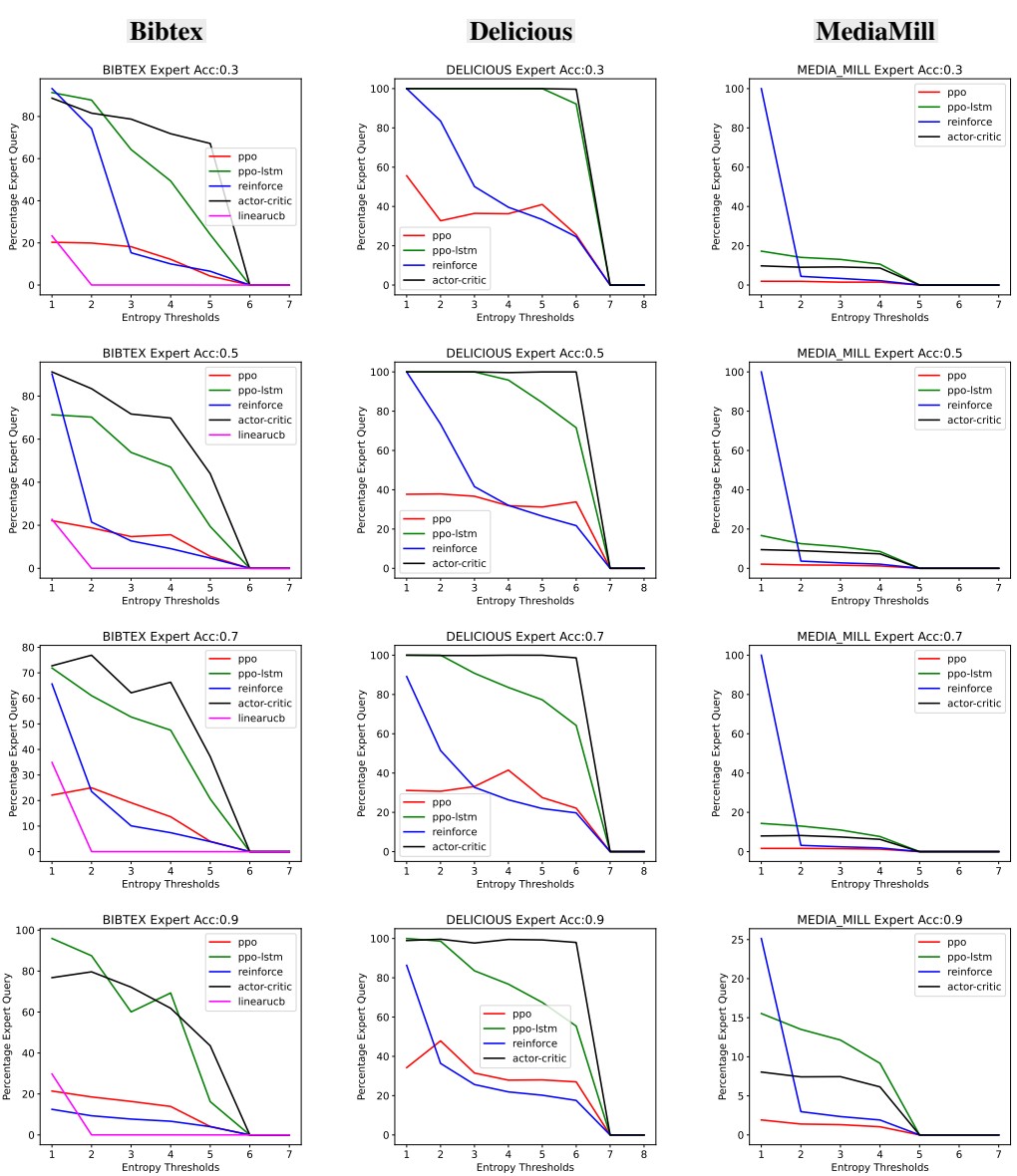

918
919
920
921
922
923
924
925
926
927
928
929
930
931
932
933
934
935
936
937
938
939
940
941
942
943
944
945
946
947
948
949
950
951
952
953
954
955
956
957
958
959
960
961
962
963
964
965
966
967
968
969
970
971

Figure 9: Variation of expert queries made for different models based on entropy for feedback type: Reward Manipulation

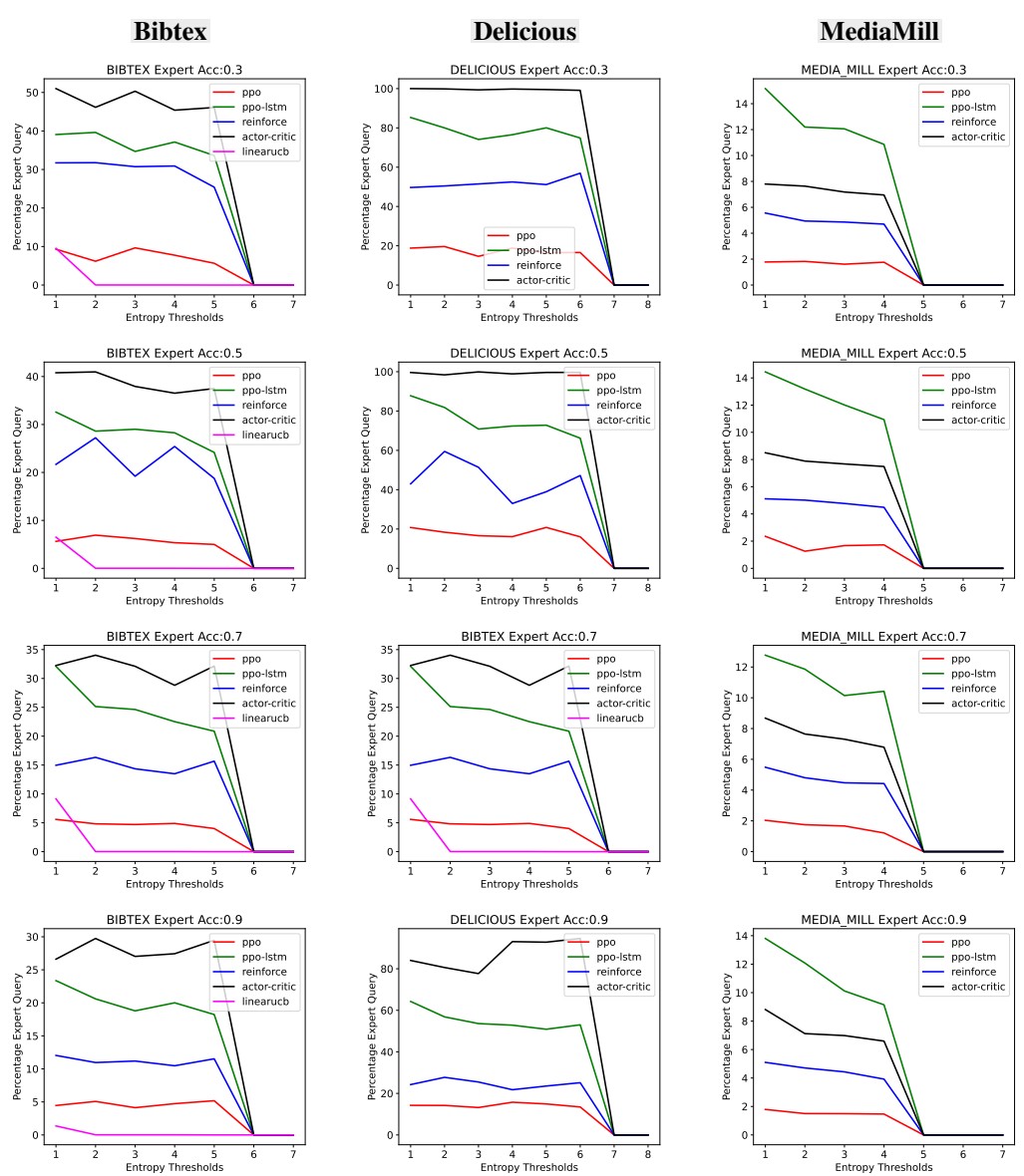

Table 2: Performance comparison of algorithms for different quality of expert feedback. The values in bold represent the maximum mean cumulative reward achieved across different levels of expert.

| Feedback Type | Algorithm Name | Environment Name | 0.3 | 0.5 | 0.7 | 0.9 |
|---|---|---|---|---|---|---|
| Action Recommendation | PPO | Bibtex | **0.27349 ± 0.00167** | 0.26383 ± 0.00091 | 0.20268 ± 0.00260 | 0.16763 ± 0.00092 |
| Reward Manipulation | PPO | Bibtex | 0.27827 ± 0.00312 | 0.27470 ± 0.00165 | 0.16965 ± 0.00202 | **0.31021 ± 0.00278** |
| Action Recommendation | PPO | Media_Mill | 0.76862 ± 0.00137 | 0.76842 ± 0.00230 | **0.77206 ± 0.00124** | 0.76662 ± 0.00134 |
| Reward Manipulation | PPO | Media_Mill | 0.76683 ± 0.00190 | 0.76530 ± 0.00128 | 0.76895 ± 0.00291 | **0.77545 ± 0.00151** |
| Action Recommendation | PPO | Delicious | **0.51770 ± 0.00220** | 0.36824 ± 0.00191 | 0.37114 ± 0.00208 | 0.46170 ± 0.00130 |
| Reward Manipulation | PPO | Delicious | **0.48187 ± 0.00113** | 0.29682 ± 0.00230 | 0.36717 ± 0.00215 | 0.40190 ± 0.00165 |
| Action Recommendation | PPO-LSTM | Bibtex | **0.13464 ± 0.00086** | 0.11283 ± 0.00204 | 0.11533 ± 0.00090 | 0.02363 ± 0.00063 |
| Reward Manipulation | PPO-LSTM | Bibtex | **0.14413 ± 0.00052** | 0.14157 ± 0.00186 | 0.13750 ± 0.00095 | 0.14304 ± 0.00136 |
| Action Recommendation | PPO-LSTM | Media_Mill | 0.76836 ± 0.00155 | 0.77318 ± 0.00141 | **0.77504 ± 0.00058** | 0.77113 ± 0.00120 |
| Reward Manipulation | PPO-LSTM | Media_Mill | 0.76973 ± 0.00114 | **0.77447 ± 0.00177** | 0.76748 ± 0.00187 | 0.76197 ± 0.00373 |
| Action Recommendation | PPO-LSTM | Delicious | **0.12497 ± 0.00140** | 0.11567 ± 0.00091 | 0.11793 ± 0.00203 | 0.11698 ± 0.00100 |
| Reward Manipulation | PPO-LSTM | Delicious | **0.28802 ± 0.00123** | 0.26663 ± 0.00204 | 0.09600 ± 0.00092 | 0.26014 ± 0.00151 |
| Action Recommendation | Reinforce | Bibtex | 0.24346 ± 0.00128 | 0.27678 ± 0.00159 | **0.29793 ± 0.00134** | 0.11714 ± 0.00133 |
| Reward Manipulation | Reinforce | Bibtex | 0.21970 ± 0.00090 | 0.24939 ± 0.00148 | 0.25543 ± 0.00166 | **0.25662 ± 0.00137** |
| Action Recommendation | Reinforce | Media_Mill | 0.08715 ± 0.00139 | 0.35710 ± 0.00214 | 0.63323 ± 0.00296 | **0.63446 ± 0.00155** |
| Reward Manipulation | Reinforce | Media_Mill | 0.77292 ± 0.00310 | 0.77098 ± 0.00177 | 0.77183 ± 0.00111 | **0.77339 ± 0.00129** |
| Action Recommendation | Reinforce | Delicious | **0.37394 ± 0.00165** | 0.35349 ± 0.00121 | 0.37268 ± 0.00230 | 0.24432 ± 0.00258 |
| Reward Manipulation | Reinforce | Delicious | 0.04502 ± 0.00067 | **0.15057 ± 0.00138** | 0.07441 ± 0.00142 | 0.07983 ± 0.00091 |
| Action Recommendation | Actor-Critic | Bibtex | 0.14119 ± 0.00107 | 0.21240 ± 0.00068 | **0.23825 ± 0.00093** | 0.15231 ± 0.00208 |
| Reward Manipulation | Actor-Critic | Bibtex | 0.17242 ± 0.00126 | **0.23110 ± 0.00149** | 0.19961 ± 0.00119 | 0.19822 ± 0.00149 |
| Action Recommendation | Actor-Critic | Media_Mill | 0.76394 ± 0.00118 | **0.77449 ± 0.00242** | 0.76325 ± 0.00085 | 0.76966 ± 0.00076 |
| Reward Manipulation | Actor-Critic | Media_Mill | 0.76749 ± 0.00205 | 0.77507 ± 0.00124 | **0.77664 ± 0.00099** | 0.76347 ± 0.00203 |
| Action Recommendation | Actor-Critic | Delicious | 0.02017 ± 0.00084 | 0.02213 ± 0.00031 | 0.02629 ± 0.00054 | **0.03498 ± 0.00036** |
| Reward Manipulation | Actor-Critic | Delicious | 0.02292 ± 0.00051 | 0.02334 ± 0.00070 | **0.02354 ± 0.00034** | 0.02154 ± 0.00069 |
| Action Recommendation | LinearUCB | Bibtex | **0.02478 ± 0.00068** | 0.02280 ± 0.00056 | 0.02145 ± 0.00066 | 0.02002 ± 0.00055 |
| Reward Manipulation | LinearUCB | Bibtex | 0.02369 ± 0.00080 | 0.02532 ± 0.00079 | 0.02518 ± 0.00049 | **0.03527 ± 0.00115** |
| Action Recommendation | LinearUCB | Media_Mill | 0.00321 ± 0.00028 | 0.00259 ± 0.00029 | **0.17961 ± 0.00117** | 0.17399 ± 0.00084 |
| Reward Manipulation | LinearUCB | Media_Mill | 0.00059 ± 0.00004 | 0.00058 ± 0.00007 | **0.19890 ± 0.00087** | 0.05337 ± 0.00136 |
| Action Recommendation | LinearUCB | Delicious | **0.02430 ± 0.00053** | 0.01818 ± 0.00036 | 0.02064 ± 0.00061 | 0.05308 ± 0.00066 |
| Reward Manipulation | LinearUCB | Delicious | 0.01664 ± 0.00022 | **0.10018 ± 0.00161** | 0.01889 ± 0.00051 | 0.08540 ± 0.00063 |
| Action Recommendation | Bootstrapped-TS | Bibtex | 0.22537 ± 0.00196 | 0.19911 ± 0.00105 | 0.21668 ± 0.00144 | **0.24097 ± 0.00137** |
| Reward Manipulation | Bootstrapped-TS | Bibtex | 0.15276 ± 0.00101 | **0.27697 ± 0.00103** | 0.18423 ± 0.00087 | 0.18468 ± 0.00278 |
| Action Recommendation | Bootstrapped-TS | Media_Mill | 0.00000 ± 0.00000 | 0.00000 ± 0.00000 | 0.00000 ± 0.00000 | 0.00000 ± 0.00000 |
| Reward Manipulation | Bootstrapped-TS | Media_Mill | 0.00000 ± 0.00000 | 0.00000 ± 0.00000 | 0.00000 ± 0.00000 | 0.00000 ± 0.00000 |
| Action Recommendation | Bootstrapped-TS | Delicious | 0.00000 ± 0.00000 | 0.00000 ± 0.00000 | 0.00000 ± 0.00000 | 0.00000 ± 0.00000 |
| Reward Manipulation | Bootstrapped-TS | Delicious | 0.00000 ± 0.00000 | 0.00000 ± 0.00000 | 0.00000 ± 0.00000 | 0.00000 ± 0.00000 |

# E  HYPER PARAMETERS

We provide the hyperparameters for the policy based RL algorithms and the range of values of entropy thresholds that we consider for each dataset.

## E.1  HYPERPARAMETERS FOR POLICY BASED RL ALGORITHMS

Table 3: HyperParameters for Policy based Algorithms. AFD=Advantage function discount.

| Algorithms | Training Epochs | Learning Rate | AFD | Clipping | Batch Size |
|---|---|---|---|---|---|
| PPO | 5000 | 0.005 | 0.1 | 0.1 | 32 |
| PPO-LSTM | 5000 | 0.001 | 0.95 | 0.1 | 32 |
| Reinforce | 5000 | 0.0002 | - | - | - |
| Actor Critic | 5000 | 0.002 | - | - | 32 |

## E.2 RANGE OF ENTROPY THRESHOLDS CONSIDERED

Table 4: Entropy thresholds for different environments $\lambda$

| Item | $\lambda$ values |
|------|------------------|
| Bibtex | 2.5, 3.5, 5.0, 6.5, 9.0 |
| Media Mill | 1.5, 2.5, 3.0, 4.5, 7.0 |
| Delicious | 1.5, 2.5, 4.5, 6.5, 9.0 |
| Yahoo | 1.5, 2.5, 4.5, 7.0, 9.0 |

## F REGRET BOUND FOR CONTEXUAL BANDITS WITH ENTROPY-BASED HUMAN FEEDBACK

Here's a regret bound for our proposed algorithm, focusing on entropy-based human feedback in a contextual bandit setting. The goal is to show how incorporating selective oracle feedback affects cumulative regret.

Let $T$ be the total number of rounds, and $\mathcal{A}$ the set of available actions. At each round $t$:

- The agent observes a context $s_t$.

- The agent selects an action $a_t \in \mathcal{A}$ based on its policy $\pi_t$, which incorporates feedback if requested.

- The oracle feedback is solicited when the entropy of the policy $H(\pi_t)$ exceeds a threshold $\lambda$.

- The observed reward $r_t(a_t)$ is a combination of environment and feedback rewards.

The expected regret at time $t$ is defined as:

$$\text{Regret}_t = \mathbb{E}[r_t(a_t^*) - r_t(a_t)],$$

where $a_t^* = \arg\max_{a \in \mathcal{A}} \mathbb{E}[r_t(a)]$ is the optimal action.

The total regret over $T$ rounds is:

$$\text{Regret}(T) = \sum_{t=1}^{T} \text{Regret}_t.$$

### F.1 THEOREM: REGRET BOUND

**Assume**: The entropy threshold $\lambda$ ensures that feedback is solicited with probability $P(H(\pi_t) > \lambda) = p$. Oracle feedback provides correct information with probability $q_t$.

Then, the expected regret of the proposed algorithm is bounded by:

$$\mathbb{E}[\text{Regret}(T)] \leq O\left(\sqrt{T|\mathcal{A}| \log T}\right) + O\left(\frac{(1-p)T}{q_t}\right).$$

1. Regret Decomposition: Decompose regret into two components:

$$\text{Regret}(T) = \sum_{t \in \mathcal{F}} \text{Regret}_t + \sum_{t \notin \mathcal{F}} \text{Regret}_t,$$

where $\mathcal{F}$ is the set of rounds where feedback is requested ($H(\pi_t) > \lambda$).

2. Regret Without Feedback ($t \notin \mathcal{F}$): When no feedback is requested, the regret follows standard contextual bandit regret:

$$\mathbb{E}[\text{Regret}_{\text{no-feedback}}(T)] \leq O(\sqrt{T|\mathcal{A}| \log T}).$$

3. Regret With Feedback ($t \in \mathcal{F}$): For rounds where feedback is solicited: (i) Feedback improves decision quality, reducing regret proportional to feedback accuracy $q_t$. (ii) The regret in feedback

rounds is bounded by $(1 - q_t)$ per round:

$$\mathbb{E}[\text{Regret}_{\text{feedback}}(T)] \leq O\left(\frac{(1 - p)T}{q_t}\right).$$

4. Combining Terms: Combining both terms yields the total regret bound:

$$\mathbb{E}[\text{Regret}(T)] \leq O\left(\sqrt{T|\mathcal{A}|\log T}\right) + O\left(\frac{(1 - p)T}{q_t}\right).$$

We have the following implications:

- Feedback Benefit: The bound highlights how oracle feedback reduces regret by improving decision-making in high-uncertainty rounds.

- Trade-off: The second term reflects the cost-benefit trade-off of feedback. With frequent and accurate feedback ($p \to 1$ and $q_t \to 1$), the regret decreases significantly.

- Entropy Threshold: The choice of $\lambda$ (affecting $p$) allows control over feedback frequency, balancing feedback cost and regret reduction.

# G   TRADE-OFFS BETWEEN ACTION RECOMMENDATION AND REWARD MANIPULATION USING LOWER BOUNDS

We can incorporate a lower bound analysis to compare the trade-offs between Action Recommendation (AR) and Reward Manipulation (RM). It highlights the theoretical benefits and limitations of each feedback type.

**Problem Setup and Notation**

Let: $T$: Total number of rounds. $K$: Number of actions. $\mathcal{A}$: Action space. $s_t$: Context observed at round $t$. $r_t(a)$: Reward for action $a$ at round $t$. $q_t^{AR}$: Probability that the feedback in AR is correct (expert recommendation quality). $q_t^{RM}$: Probability that the reward signal is correctly modified in RM (expert reward quality). $p_t$: Probability of querying feedback in either AR or RM.

We aim to derive regret lower bounds for both feedback types and analyze their trade-offs.

## G.1   ACTION RECOMMENDATION (AR)

In the AR setting: The agent queries the oracle to receive the recommended action $a_t^{AR}$, which is assumed to be correct with probability $q_t^{AR}$.

**Regret Lower Bound for AR**

In rounds where feedback is not queried $(1 - p_t)$, the regret follows standard contextual bandit bounds:

$$\mathbb{E}[\text{Regret}_{\text{no-feedback}}(T)] \geq O((1 - p_t)\sqrt{TK}).$$

In rounds where AR feedback is queried $(p_t)$, regret depends on the quality of the recommended action:

$$\mathbb{E}[\text{Regret}_{\text{feedback}}^{AR}(T)] \geq O\left(\frac{p_t T}{q_t^{AR}}\right).$$

Thus, the total regret for AR is bounded by:

$$\mathbb{E}[\text{Regret}^{AR}(T)] \geq O((1 - p_t)\sqrt{TK}) + O\left(\frac{p_t T}{q_t^{AR}}\right).$$

## G.2   REWARD MANIPULATION (RM)

In the RM setting: The agent receives a modified reward signal $\tilde{r}_t(a_t)$, adjusted by the oracle to reflect feedback quality $q_t^{RM}$.

**Regret Lower Bound for RM**

Without feedback $(1 - p_t)$:

$$\mathbb{E}[\text{Regret}_{\text{no-feedback}}(T)] \geq O((1 - p_t)\sqrt{TK}).$$

With RM feedback $(p_t)$, the manipulated reward provides improved reward estimates, reducing regret:

$$\mathbb{E}[\text{Regret}_{\text{feedback}}^{RM}(T)] \geq O\left(\frac{p_t T}{q_t^{RM}}\right).$$

Thus, the total regret for RM is:

$$\mathbb{E}[\text{Regret}^{RM}(T)] \geq O((1 - p_t)\sqrt{TK}) + O\left(\frac{p_t T}{q_t^{RM}}\right).$$

### G.3 TRADE-OFF ANALYSIS

**1. Feedback Quality** $q_t^{AR}$ **vs.** $q_t^{RM}$: **AR** directly impacts action selection, which may lead to larger regret reduction if $q_t^{AR}$ is high. **RM** improves the reward signal, which may be less direct but still effective in guiding future decisions.

**2. Feedback Frequency** $p_t$: Both AR and RM benefit from higher feedback frequency $p_t$. However, querying feedback comes with costs, and the choice depends on the relative quality of feedback $q_t$.

**3. Cumulative Regret**: If $q_t^{AR} > q_t^{RM}$, AR is more effective in reducing regret:

$$\mathbb{E}[\text{Regret}^{AR}(T)] < \mathbb{E}[\text{Regret}^{RM}(T)].$$

Conversely, if $q_t^{RM}$ is higher, RM could achieve lower regret.

### G.4 PRACTICAL IMPLICATIONS

**When to Use AR**: (i) When action recommendations are highly reliable ($q_t^{AR} \to 1$). (ii) When immediate corrective feedback on actions is critical.

**When to Use RM**: (i) When action recommendations are less reliable, but reward signals can be improved consistently ($q_t^{RM} > q_t^{AR}$). (ii) When reward shaping can better guide learning in uncertain environments.

This analysis shows that the choice between AR and RM depends on the quality and frequency of feedback. Both methods have distinct strengths, and their trade-offs can be quantified using the derived regret bounds. Future work could further explore hybrid strategies that dynamically balance AR and RM based on real-time feedback quality.

## H DETAILED ANALYSIS OF FEEDBACK SOLICITATION COSTS AND THEIR IMPACT ON CUMULATIVE REWARDS

In systems that integrate human feedback, the cost of feedback solicitation plays a crucial role in determining the efficiency and practicality of the algorithm. Below, we provide a structured analysis of these costs and their effects.

### H.1 COST COMPONENTS IN FEEDBACK SOLICITATION

Feedback solicitation costs can be broken into three primary components:

- **Human Effort Cost** ($C_h$): Time, cognitive load, or financial compensation required for a human expert to provide feedback.
- **System Overhead** ($C_s$): Computational and communication overhead associated with querying, collecting, and processing feedback.

- **Opportunity Cost** ($C_o$): Delay or missed opportunities to explore other actions during feedback solicitation.

The total cost per solicitation can be expressed as:

$$C_{\text{total}} = C_h + C_s + C_o.$$

## H.2 Trade-off Between Feedback and Performance

Feedback improves learning by reducing uncertainty in decision-making but comes at a cost. The trade-off is evident in two opposing factors:

- **Benefits**: Incorporating feedback accelerates convergence, reduces regret, and improves cumulative rewards.
- **Costs**: Frequent feedback queries increase the total cost, potentially diminishing the system's overall utility.

The cumulative rewards $R_T$ after $T$ rounds with feedback solicitation frequency $p$ can be modeled as:

$$R_T = \sum_{t=1}^{T} r_t - p \cdot C_{\text{total}},$$

where $r_t$ represents the reward at time step $t$, and $p$ is the fraction of rounds in which feedback is solicited.

## H.3 Effect of Feedback Quality and Frequency

### H.3.1 High-Quality Feedback ($q_t \rightarrow 1$)

- **Impact**: High-quality feedback significantly reduces regret, as the system quickly learns optimal actions.
- **Cost Justification**: Even with higher solicitation costs, the performance gains justify frequent feedback, especially in complex environments.

### H.3.2 Low-Quality Feedback ($q_t \rightarrow 0$)

- **Impact**: Low-quality feedback adds noise to the learning process, diminishing performance gains.
- **Cost Justification**: Frequent solicitation becomes inefficient, and selective feedback solicitation based on entropy thresholds ($\lambda$) is preferred.

### H.3.3 Frequency of Feedback ($p$)

- High $p$ improves learning but incurs higher total costs, leading to diminishing returns as cumulative rewards plateau.
- Low $p$ reduces costs but risks slower convergence and higher regret.

## H.4 Entropy-Based Feedback Solicitation

An entropy-based mechanism optimizes feedback solicitation by querying only when the model's uncertainty surpasses a predefined threshold ($\lambda$):

- **High Entropy** ($H(\pi) > \lambda$): Feedback is requested to resolve uncertainty, ensuring maximum utility from the cost incurred.
- **Low Entropy** ($H(\pi) \leq \lambda$): Feedback is avoided as the model is confident in its decision.

This selective querying strategy reduces the total feedback cost while maintaining performance by focusing resources where they have the highest impact.

## H.5 EXPERIMENTAL ANALYSIS

Using simulated environments:

- **Performance vs. Cost**: Reducing feedback frequency ($p$) by increasing $\lambda$ leads to a marginal decrease in performance while significantly reducing costs. For instance, at $p = 0.3$, performance dropped by only 5% compared to $p = 1.0$, but the cost was reduced by 70%.

- **Dataset Dependency**: Feedback efficiency varies across datasets. Datasets with large action spaces benefit more from frequent feedback (e.g., Delicious dataset), while datasets with fewer actions (e.g., Bibtex dataset) require less frequent feedback due to faster convergence.

## H.6 INSIGHTS AND PRACTICAL IMPLICATIONS

- **Optimal Feedback Strategy**: Use selective feedback based on model uncertainty and adjust $\lambda$ to balance feedback costs with performance gains depending on the application.

- **Recommendations for Practitioners**: In high-cost settings, prioritize low feedback frequency ($p \rightarrow 0.2 - 0.4$) with robust entropy thresholds. For critical applications, higher feedback costs can be justified for improved cumulative rewards.

- **Scalability**: Entropy-based solicitation is particularly effective for large-scale systems where querying all rounds is impractical.

## H.7 CONCLUSION

Balancing feedback solicitation costs and cumulative rewards requires careful tuning of feedback frequency and quality thresholds. An entropy-based approach effectively minimizes costs while maintaining performance, making it a practical solution for real-world applications. Future work could explore dynamic threshold adaptation to further optimize this trade-off.

