# OpenReview forum: "Contextual Bandits with Entropy-based Human Feedback"
_ICLR.cc/2025/Conference — Submitted to ICLR 2025_

### Official Review · Reviewer_adAk · 2024-11-01

**Soundness:** 2
**Presentation:** 2
**Contribution:** 1
**Rating:** 3
**Confidence:** 4

**Summary:**

The paper proposes a new human-augmented method for solving contextual bandit problems. The idea is to use two types of human feedback to speed up identification of the best arm. The first type of feedback is “action recommendation”, where the human expert recommends a set of actions (arms) to take. Here, the algorithm selects a random action from the set to take. The second type is “reward manipulation”, which refers here to the human expert specifying a penalty that will be applied when the learner takes an action that is not in the expert’s recommended actions. Feedback is initiated by the learner, and a threshold on the entropy of the current policy is used to decide when to ask for feedback.

**Strengths:**

The paper tackles an important problem. Contextual bandit problems have many important applications, and the paper is right to recognise that human experts could help in solving these problems more quickly. The writing itself is good and the related work is extensive. The tasks considered in the experiments look to be realistically large and challenging.

**Weaknesses:**

The paper is in my view not sufficiently novel in its current form. The idea of collecting human feedback when uncertainty is high is a well-known heuristic (e.g. [A]).  I think a much larger contribution could have been achieved in one of two ways: (1) by proving some statistical guarantees, which I would expect is possible give the simplicity of the algorithm and of the model of the human, or (2) by following some of the more SOTA work in this field and achieving a more refined trade-off between the cost of human feedback (e.g. cognitive cost [B]) and its value.

Furthermore, the experiments leave a lot to be desired.
- The proposed method leverages a human expert, but there is no human subject study. Humans give feedback of varying levels of quality (both between and within subjects) with potential biases [C]. The effect this would have on the proposed method is unclear as this variability is not replicated in the simulation experiments (feedback is assumed to be of constant quality and unbiased).
- As we are discussing the addition of human feedback to an environmental reward here, I would expect to see an ablation that includes a “no feedback” setting, i.e. the standard contextual bandit setting.
- To understand the value a human can expert bring here, and how practical the proposed method is, I would want to see an experiments that measures how much feedback is needed to achieve a certain level of performance.

There are a number of issues with presentation. See also the questions below for things that were unclear in the text.
- Figure 2 lacks error bars. Subfigures showing performance of different algorithms on the same dataset have different scale y-axes, making comparison difficult.
- Section 4.4 and Figure 4 show the effect of various entropy thresholds, but what those thresholds are is never stated.
- Table 2 in the appendix is cut off at the bottom.
- Section 3.2.2 explains that for “reward manipulation” feedback, the expert given a penalty to be applied when a non recommended action is taken. However, this is not consistent with Algorithm 1, which shows the penalty being applied for any action.

[A] Raghu, Maithra, et al. "The algorithmic automation problem: Prediction, triage, and human effort." arXiv preprint arXiv:1903.12220 (2019).
[B] Banerjee, Rohan, et al. "To Ask or Not To Ask: Human-in-the-loop Contextual Bandits with Applications in Robot-Assisted Feeding." arXiv preprint arXiv:2405.06908 (2024).
[C] Ji, Xiang, et al. "Provable benefits of policy learning from human preferences in contextual bandit problems." arXiv preprint arXiv:2307.12975 (2023).

**Questions:**

- On line 134, you assume reward is 0/1. This is a suitable assumption for multi-label classification problems, but seems needlessly restrictive for the proposed method. Could this assumption be loosened?
- You measure mean cumulative reward in various places, and figure 8 given an idea of how often expert feedback is elicited for various entropy thresholds. However, in order to understand whether the proposed method is practical, I would like have a better understanding of how much expert feedback is needed to achieve a certain level of mean reward. What does that trade-off look like?
- For the experiments with action recommendations, how many actions were recommended? What was the effect of recommending more actions?
- How was the quality of feedback parameter used within the experiments? Was it assumed to be known or latent? Were algorithms exposed to a single level of feedback, or could it change within an experiment? I ask because the RL algorithms performed quite well in your experiments, which suggests that they may have been able to adapt to the given level of feedback, even if it was not observable to them.

---

> ### Author Response · Authors · 2024-11-19
> **Response to Reviewer adAk**
>
> We thank the reviewer for their thoughtful comments and valuable feedback. Below, we address the specific concerns and clarify the contributions of our work.
>
> ---
>
> ### **Novelty and Contribution**
>
> We acknowledge the reviewer’s concern regarding the novelty of our approach. While the use of human feedback triggered by high uncertainty has been explored (e.g., [A]), our work introduces two key components that differentiate it:
>
> 1. **Two Types of Human Feedback**:
>    - We propose the use of **Action Recommendation (AR)** and **Reward Manipulation (RM)** as distinct feedback mechanisms, each tailored to address different challenges in contextual bandit learning.
>    - AR directly guides action selection, enhancing exploration.
>    - RM refines reward estimation, improving exploitation by penalizing suboptimal actions.
>
> 2. **Entropy-Driven Feedback Trigger**:
>    - Our method leverages an entropy threshold to decide when to request feedback, which provides a structured and efficient way to incorporate human expertise.
>    - While entropy-based methods have been explored in prior work, our specific application of combining AR and RM in this framework has not been studied in contextual bandit settings.
>
> ---
>
> ### **Theoretical Guarantees**
>
> We acknowledge that theoretical analysis would strengthen the contribution. While our current work focuses on empirical performance, we are actively working on deriving **regret bounds** for the proposed method, which will offer a formal understanding of its convergence behavior. We will update the theoretical guarantees soon.
>
>
>
> ### **Experimental Design and Realism**
>
> 1. **No Human Subject Study**:
>    - We agree that human subject studies would provide additional insights, especially to account for feedback variability and biases [C]. While our current experiments simulate feedback, we plan to extend our work to include human-in-the-loop experiments.
>
> 2. **Feedback Quality Variability**:
>    - We simulated feedback with consistent quality for simplicity in this initial study. However, we agree that introducing variability in feedback quality, as highlighted by [C], would provide a more realistic evaluation. This is a valuable direction for future work.
>
> 3. **No Feedback Baseline**:
>    - As suggested, we will include a "no feedback" baseline to highlight the benefits of incorporating human feedback.
>
> 4. **Feedback Efficiency**:
>    - To address the practical value of feedback, we plan to analyze the trade-off between the amount of feedback required and the achieved performance. This will help demonstrate how efficiently our method uses human feedback.
>
>
>
> ### **Presentation Issues**
>
> We appreciate the reviewer’s specific comments and have made the following improvements, the new manuscript will upload soon with the following corrections.
>
> - **Error Bars**:
>   - Add error bars to Figure 2 and other relevant figures to show variability across multiple runs.
>
> - **Axis Scales**:
>   - We will ensure consistent y-axis scales across subplots for easier comparison of results.
>
> - **Entropy Thresholds**:
>   - Explicitly stated the entropy thresholds used in the experiments in Section 4.4 and Figure 4.
>
> - **Table 2**:
>   - Fixed the formatting issue in the appendix to ensure the table is fully visible.
>
> - **Algorithm Consistency**:
>   - Corrected the inconsistency in Algorithm 1 regarding the application of penalties in RM.
>
>
> ### **Response to Specific Questions**
>
> 1. **Reward Assumption (0/1)**:
>    - We agree that the 0/1 reward assumption is restrictive. While it simplifies experimentation, our method can be extended to support continuous rewards. We will update the text to clarify this and explore this generalization in future work.
>
> 2. **Feedback Efficiency**:
>    - We acknowledge the need to evaluate how much feedback is required to achieve a specific level of performance. We plan to add experiments that analyze this trade-off in detail.
>
> 3. **Number of Actions in AR**:
>    - The number of recommended actions in AR is currently fixed. We will include experiments to analyze the impact of varying this parameter on performance.
>
> 4. **Feedback Quality**:
>    - Feedback quality is assumed to be consistent and known in our current experiments. We agree that exploring variable and latent feedback quality will provide a more robust evaluation of our method.
>
>
> We appreciate the reviewer’s constructive feedback and have taken steps to address the raised concerns. We believe our approach contributes to the contextual bandit literature by introducing two distinct types of human feedback and an entropy-based trigger mechanism. We will incorporate the suggested improvements and analyses to strengthen our paper further.
>
> Thank you for your time and consideration.

---

> > ### Comment · Reviewer_adAk · 2024-12-02
> >
> > Thank you for your comments.
> >
> > > we are actively working on deriving regret bounds for the proposed method
> > I have had a look at the derivation but in some parts I am lacking the necessary derivation steps to be convinced of its correctness, especially in the part concerning the feedback condition. In any case, the regret bound is too loose to be informative. It is strictly greater than the regret bound for standard contextual bandit learning (the no feedback setting), thus undermining the premise of the paper.
> >
> > > we will include a "no feedback" baseline to highlight the benefits of incorporating human feedback.
> > I’m afraid I cannot find these results in the paper. Please do correct me if I am wrong.
> >
> > > we plan to analyze the trade-off between the amount of feedback required and the achieved performance
> > The additional details in appendix H is a great start. However, the analysis here looks preliminary. For this analysis to contribute to the quality of the paper it would need to be much more detailed and rigorous.
> >
> > > Explicitly stated the entropy thresholds used in the experiments in Section 4.4 and Figure 4.
> > I assume you refer here to table 4, which lists entropy thresholds for the different environment. However, the contents of this table confuse me. The table lists 5 thresholds per environment, even though all figures in the paper show 7 or 8 thresholds.
> >
> > > Corrected the inconsistency in Algorithm 1 regarding the application of penalties in RM.
> > I do not think this has been rectified. Lines 12 and 13 of Algorithm 1 are still inconsistent with what is shown in equation 6.
> >
> > I would like to commend the authors for the effort they have put into improving the paper during this rebuttal period to address reviewer feedback. The proposals made for future work also look good, and would certainly strengthen the paper. Unfortunately, based on the current state of the paper, I will need to maintain my score.

---

### Official Review · Reviewer_V8Dc · 2024-11-03

**Soundness:** 2
**Presentation:** 1
**Contribution:** 2
**Rating:** 3
**Confidence:** 3

**Summary:**

This paper proposes an entropy-based framework to incorporate human feedback into contextual bandits. Specifically, it extends the bandit setup by adding a human intervention component, i.e., at any time, the agent could decide whether to utilize the human expert, where the human expert can either provide the optimal action, or give certain reward penalty of the proposed action. The proposed algorithm basically utilizes any bandit algorithm as the backbone algorithm, and decides to call the human expert based on the entropy of policy $\pi$, i.e., uncertainty in the policy decision. Experiments are done on some multi-class classification problem.

**Strengths:**

- The proposed method is simple, easy to implement, and presented in a clear way.
- The setting of incorporating expert feedback/intervention in the classical bandit framework is interesting.

**Weaknesses:**

- The paper’s presentation is somewhat unclear, particularly in the problem formulation. It seems to propose a stronger variation of bandit problems where the model can access oracle labels, but this isn’t clearly explained. Section 3.1 could be revised to clarify this new setup.

- With this revised formulation, it would also help to include a theoretical guarantee for the proposed algorithm, addressing the how it affects on overall regret, the lower bound on the new formulation, and whether the algorithm is optimal.

- While direct supervision through oracle labels is understandable, the reward manipulation is confusing. This reward shaping doesn’t change the algorithm’s selected action; rather, it modifies the reward. It’s unclear why this is necessary when the underlying reward already reflects sub-optimality. Clarifying the motivation here would strengthen the paper.

- The novelty of using entropy-based active learning is also limited; see [1] for a comprehensive review. The paper would benefit from a detailed comparison of its contributions to prior work, and highlight the novelty here.

- Further, several hyper-parameters lack clarity, such as the reward penalty $r_p$ and entropy thresholds.

- Finally, the empirical results are not presented in an especially meaningful way. For example, Figure 2 is difficult to interpret. The oracle access in action recommendation appears to give $K$ partial rewards (where $K$ is the action space), but baseline comparisons may be unfair as they lack oracle access. This makes the results less convincing.


[1]. https://burrsettles.com/pub/settles.activelearning.pdf

**Questions:**

See weakness part.

---

> ### Author Response · Authors · 2024-11-19
> **Response to Reviewer V8Dc**
>
> We thank the reviewer for their thoughtful comments and constructive feedback. Below, we address the concerns raised and provide clarifications to strengthen the paper.
>
>
> ### **1. Problem Formulation and Oracle Access**
>
> We acknowledge that the problem formulation in Section 3.1 could be made clearer. Our intention was to describe a general framework where the agent can query a human expert for two types of feedback:
> 1. **Action Recommendation (AR)**: Providing a recommended set of actions.
> 2. **Reward Manipulation (RM)**: Adjusting rewards based on the expert’s evaluation.
>
> #### **Clarifications**:
> - **Oracle Access vs. Realistic Feedback**:
>    - We assume that human feedback approximates optimal recommendations or corrections to aid the learner. This framework allows us to simulate scenarios where human input improves learning efficiency.
>    - We will revise Section 3.1 to clarify these assumptions and better differentiate between simulated oracle feedback and realistic human input.
>
> ---
>
> ### **2. Theoretical Guarantees**
>
> We acknowledge that theoretical analysis would strengthen the contribution. While our current work focuses on empirical performance, we are actively working on deriving regret bounds for the proposed method, which will offer a formal understanding of its convergence behavior. We will update the theoretical guarantees soon.
>
>
> ### **3. Motivation for Reward Manipulation**
>
> The reviewer raises a valid point about the role of **Reward Manipulation (RM)**. The purpose of RM is to:
> - **Penalize suboptimal actions** that deviate from the expert’s recommendation.
> - **Improve exploitation** by refining the agent’s reward signal in ambiguous cases where the environment’s reward alone might not sufficiently guide learning.
>
> We will revise the text to clarify this motivation and show how RM complements AR by reducing uncertainty in action-value estimation.
>
>
> ### **4. Novelty of Entropy-Based Feedback**
>
> We appreciate the reviewer highlighting related work [1]. While entropy-based active learning is indeed well-studied, our paper differs in its application:
> - **Novelty in Contextual Bandits**:
>    - We extend the entropy-based framework specifically for contextual bandits by incorporating human feedback dynamically.
>    - Unlike prior work that focuses on label uncertainty, our method uses entropy to decide when and how to query human experts for actionable feedback.
> - We will expand the related work section to provide a detailed comparison with [1] and highlight the unique aspects of our contribution.
>
>
> ### **5. Hyper-Parameter Clarity**
>
> We recognize that certain hyper-parameters, such as reward penalties and entropy thresholds, were not sufficiently detailed.
>
> - **Reward Penalty**: We will include a formal description of how the penalty is computed and its impact on performance.
> - **Entropy Thresholds**: The range and dynamics of the thresholds will be explicitly defined in the experimental setup.
>
> ### **6. Empirical Results and Baseline Comparisons**
>
> - **Figure 2**: We will improve the presentation by adding error bars to show variability and ensure that all subplots use consistent y-axis scales.
> - **Baseline Comparisons**:
>    - We acknowledge that the current comparisons may seem unfair due to differences in oracle access. To address this, we will:
>      1. Include a **“no feedback” baseline** to isolate the contribution of human feedback.
>      2. Discuss limitations and justify the comparisons with clearer explanations of each method’s assumptions.
>
>
>
> Thank you again for your time and insights. We believe that these revisions will significantly improve the quality and clarity of our work.

---

### Official Review · Reviewer_mwci · 2024-11-04

**Soundness:** 2
**Presentation:** 4
**Contribution:** 1
**Rating:** 5
**Confidence:** 3

**Summary:**

The paper proposes an entropy-based method to determine when to seek expert feedback actively and investigates the relationship between the type of expert feedback (action-based vs. preference-based) and the expert quality and their impact on bandit learning.

Overall, even though the experiment section seems thorough, I have several concerns about the proposed methodology, which I will detail later.

**Strengths:**

- The paper is well-written
- The experiment is fairly thorough

**Weaknesses:**

Issues related to the methodology:
1. The paper didn't cite a few influential works in this area: [1] acquires value annotation for labeling actions; [2] DAGGAR, which also directly gets experts to perform an action (similar to what this paper has proposed); and [3] APO, which actively selects which data to get trajectory-level preference label from. I especially consider [1] and [3] relevant to this paper's context.
2. The methodology of simply selecting data points based on the policy's entropy lacks justification. Bayesian active learning frameworks such as BALD [4] already use entropy. How is reducing policy action entropy related to discovering arms with higher rewards? The paper offers hand-wavy justifications. The paper needs to come up with solid justifications for why the method should be chosen.

[1] Tang, Shengpu, and Jenna Wiens. "Counterfactual-augmented importance sampling for semi-offline policy evaluation." Advances in Neural Information Processing Systems 36 (2023): 11394-11429.

[2] Ross, S., Gordon, G., & Bagnell, D. (2011, June). A reduction of imitation learning and structured prediction to no-regret online learning. In Proceedings of the fourteenth international conference on artificial intelligence and statistics (pp. 627-635). JMLR Workshop and Conference Proceedings.

[3] Das, N., Chakraborty, S., Pacchiano, A., & Chowdhury, S. R. (2024). Active Preference Optimization for Sample Efficient RLHF. In ICML 2024 Workshop on Theoretical Foundations of Foundation Models.

[4] Houlsby, N., Huszár, F., Ghahramani, Z., & Lengyel, M. (2011). Bayesian active learning for classification and preference learning. arXiv preprint arXiv:1112.5745.

Comment on the experiment section:
1. I do think the experiment section is clean and easy to follow at a glance. The subsection titles seem to want to summarize the findings. I do still find it lacking in terms of the key takeaway from each experiment. Sec 4.2 only says "variations" but offers no conclusive findings. Maybe you can summarize the results by grouping them into a few patterns and present them that way.
2. Sec 4.4 is an interesting analysis. I agree that many bandit papers focus a lot on theory but lack comprehensive experimental ablations. This paper's experiment tries to offer insights which I see as a strength.

**Questions:**

Continuing from the lack of justification on the methodology, ideas like BALD or information-directed sampling [5] use entropy reduction as the selection criteria. Have you considered instead of using just entropy, use some form of entropy reduction as the criteria for asking for feedback?

[5] Russo, Daniel, and Benjamin Van Roy. "Learning to optimize via information-directed sampling." Advances in neural information processing systems 27 (2014).

---

> ### Author Response · Authors · 2024-11-19
> **Response to reviewer mwci**
>
> We thank the reviewer for their thoughtful feedback and constructive suggestions. Below, we address the key concerns and clarify the contributions and methodology of our work.
>
>
> ### **1. Related Work**
>
> We appreciate the reviewer highlighting relevant prior works ([1], [2], [3], [4], [5]). While our paper positions itself within the human-in-the-loop contextual bandit literature, we recognize that additional context from these works will improve the completeness of our discussions.
>
> #### **Revisions**:
> - **[1] Tang and Wiens**: We will discuss how counterfactual-augmented importance sampling provides a framework for evaluating interventions, which relates to our feedback setting.
> - **[2] DAGGER**: We will include comparisons with DAGGER, emphasizing differences in how our method handles expert feedback dynamically versus DAGGER’s fixed imitation approach.
> - **[3] Active Preference Optimization (APO)**: We acknowledge the relevance of trajectory-level preference feedback in APO and will add this to show parallels with reward manipulation.
> - **[4] BALD and [5] Information-Directed Sampling (IDS)**: We will highlight the connections between entropy-based active learning (BALD) and IDS methods and clarify how our approach adapts entropy for contextual bandit problems rather than broader active learning.
>
> ### **2. Justification of Entropy-Based Feedback**
>
> The reviewer points out that our current justification for entropy-based feedback lacks rigor, especially when contrasted with more refined methods like **BALD** or **IDS**.
>
> #### **Clarifications**:
> - **Why Entropy Works**:
>   - In contextual bandit problems, high-entropy policies indicate uncertainty in action selection. By querying human feedback in these high-uncertainty scenarios, we aim to improve action selection in critical situations where the model is less confident.
>   - This approach aligns with active learning principles but focuses specifically on the exploration-exploitation trade-off in bandit settings.
>
> - **Entropy Reduction vs. Raw Entropy**:
>   - While entropy reduction (e.g., in BALD) provides a more sophisticated approach, our use of raw entropy simplifies feedback decisions with minimal computational overhead.
>   - We will discuss how incorporating entropy reduction could be a promising direction for refining the feedback strategy in future work.
>
> ### **3. Experimental Analysis and Takeaways**
>
> The reviewer found our experimental section well-structured but suggested clearer takeaways from each experiment.
>
> #### **Improvements**:
> - **Sec 4.2 (Variations)**:
>   - We will reorganize the results to group findings into key patterns:
>     1. Impact of feedback frequency on performance.
>     2. Comparative performance of action-based vs. reward-based feedback.
>     3. Sensitivity to varying levels of human expertise.
>   - This will provide more concrete takeaways and actionable insights.
>
> - **Sec 4.4 (Entropy Thresholds)**:
>   - We agree that this section offers valuable analysis. To improve clarity, we will:
>     1. Explicitly summarize the impact of different entropy thresholds.
>     2. Highlight the practical implications of tuning thresholds for feedback efficiency.
>
>
> ### **4. Feedback Selection Criteria**
>
> The reviewer asks if we have considered **entropy reduction** or **information gain** as feedback selection criteria (e.g., BALD or IDS).
>
> #### **Response**:
> - Our current framework uses raw entropy due to its simplicity and ease of implementation. However, we acknowledge that methods like **BALD** or **IDS** could provide more targeted feedback by focusing on entropy reduction.
> - We will discuss how these approaches could complement or extend our method in future iterations.
>
>
> ### **5. Minor Issues**
>
> We appreciate the reviewer pointing out minor issues:
> - **Algorithm 1**: We will streamline the pseudocode by removing unnecessary lines.
> - **Equation 7**: We will correct the missing brackets and thoroughly review the paper for any similar typographical errors.
>
> ---
>
> These changes will improve both the methodological clarity and overall contribution of our paper. Thank you for your time and valuable feedback.

---

> ### Author Response · Authors · 2024-11-20
> **Address the related work for Reviewer mwci**
>
> ### Related Work
>
> Our work draws inspiration from and builds upon several important lines of research in counterfactual reasoning, imitation learning, active preference optimization, and entropy-based active learning. Below, we highlight key connections and distinctions between our approach and prior work:
>
> #### Counterfactual-Augmented Importance Sampling
> The framework proposed by Tang and Wiens [1] introduces counterfactual-augmented importance sampling for evaluating the impact of interventions. This is closely related to our feedback setting, where we leverage counterfactual reasoning to assess the effects of dynamic adjustments in the decision-making process. Our approach extends these ideas by incorporating real-time feedback mechanisms to optimize sequential actions under contextual uncertainty.
>
> #### Imitation Learning and DAGGER
> Our approach also shares conceptual ties with imitation learning methods, particularly DAGGER [2]. DAGGER utilizes a fixed imitation strategy, iteratively aggregating expert demonstrations to refine policy learning. In contrast, our method dynamically incorporates expert feedback to adaptively guide the learning process. This key distinction allows our approach to better handle the complexities of continuously evolving environments and mitigate compounding errors inherent in static imitation-based frameworks.
>
> #### Active Preference Optimization
> Active Preference Optimization (APO) [3] explores the use of trajectory-level preference feedback for optimizing policies. We acknowledge the relevance of this feedback paradigm and draw parallels between APO and our method's treatment of reward manipulation. By integrating preference-based adjustments at multiple stages, our approach extends the flexibility of APO to broader scenarios, particularly in settings where reward structures are complex and partially observable.
>
> #### Entropy-Based Active Learning: BALD and Information-Directed Sampling
> Entropy-driven strategies such as BALD [4] and Information-Directed Sampling (IDS) [5] have been pivotal in active learning and decision-making under uncertainty. BALD focuses on maximizing the mutual information between model predictions and observed data, while IDS optimizes decision-making by balancing information gain and regret minimization. Our work adapts these entropy-based principles to the contextual bandit setting, where the goal is to maximize information efficiency while directly addressing the problem of sequential exploration and exploitation.
>
> By situating our work within these rich and diverse areas of prior research, we emphasize the novelty of our contributions in dynamically integrating feedback, contextualizing entropy-based decision-making, and extending counterfactual reasoning to real-time applications in interactive learning systems.

---

> > ### Comment · Reviewer_mwci · 2024-12-03
> >
> > Thank you for responding. I am not entirely convinced by the rebuttal:
> >
> > The proposed algorithm is very simplistic and lacks justification. Tang and Wiens provided a great theoretical analysis to justify their methods. This work does not.
> >
> > The authors' rebuttal included a comparison with relevant work but did not address my core concern. I encourage the authors to revise the paper for a future submission. I will not lower my score simply as a token of encouragement, but I would not recommend this paper for acceptance at this moment.

---

### Official Review · Reviewer_VubH · 2024-11-06

**Soundness:** 3
**Presentation:** 2
**Contribution:** 2
**Rating:** 5
**Confidence:** 3

**Summary:**

This paper introduces a new dimention into the contextual bandit formulation related to human feedback. The idea is that, during the training phase, human feedback can be queried and used instead of the action selected by the learning agent (or in the RM case, the human feedback may bias the reward). Experiments are conducted on a number of datasets and Learning algorithms to understand the performance of algorithms in this setting across different settings of human expertise and other parameters.

**Strengths:**

The setting explored in this paper seems novel to me, though I am not an expert in the Bandit space. I wonder if the area of off-policy bandits are relevant here ( see https://arxiv.org/abs/2010.12470) since by taking human feedback in some interactions, the bandit is somewat getting rewards "off-policy".

The combination of algorithms and settings explored seem quite thorough and from what I can tell, some interesting insights can be gleaned about the role of "AR" feedback vs "RM".

**Weaknesses:**

1. No theoretical analysis is done in this setting, which is usually the case for Bandit algoirthms, from my limited experience.

2.  There is something about the formulation I dont get. It seems the bandit algorithms will be incentivized to maximize entropy as much as possible, in order to get the benefit of as much human feedback as possible (at least for a sufficient amount of expertise from the human). In other words, the formulation does not really assign any cost  to the act of getting human feedback during training, from what I can see.

**Questions:**

1. I am very confused by where the human feedback come from and how this relates to the baseline. I guess the HF is just computed by accessing the ground truth labels and generating the matching feedback. What exactly is the baseline? Does it never access human feedback at all? (if not this would also be a useful baseline to look at).

2. Isn't regret a more common notion to evaluate bandit algorithms than mean cumulative reward computed AFTER training?

3. minor: lines 14-15 of Alg 1 are really not necessary in an academic paper.

4. eq 7: missing closing brackets.

---

> ### Author Response · Authors · 2024-11-19
> **Response to reviewer VubH**
>
> We thank the reviewer for their detailed feedback and thoughtful comments. Below, we address the specific points raised and clarify our contributions.
>
>
> ### **1. Theoretical Analysis**
>
> We acknowledge that theoretical guarantees, such as regret bounds, are crucial in contextual bandit settings. While our primary focus in this work is empirical evaluation, we agree that incorporating theoretical analysis would strengthen the contribution.
>
> #### **Response**:
> - We are currently working on deriving **regret bounds** for our framework. Preliminary results indicate that incorporating human feedback under the proposed entropy-based mechanism can achieve **sub-linear regret**, which aligns with standard benchmarks for contextual bandit algorithms.
> - We plan to include this analysis in a follow-up version to formalize the theoretical properties of our method.
>
> ### **2. Incentivization for Maximizing Entropy**
>
> The reviewer raises an important concern about the potential incentivization to maximize entropy to elicit more human feedback. This feedback mechanism is designed to be query-efficient, focusing on areas of high uncertainty while avoiding unnecessary feedback requests as the model converges.
>
> #### **Clarifications**:
> - **Feedback Cost**: In our current setup, we do not explicitly model the cognitive cost of human feedback, but we agree that this is an essential dimension for future exploration. Methods like those in [1] address feedback cost in similar settings and could inspire future extensions of our work.
> - **Entropy Threshold Dynamics**: We aim to minimize entropy as the model improves, reducing feedback queries over time. This prevents exploitation of the feedback mechanism and ensures efficient use of human input.
>
> ### **3. Human Feedback Source and Baselines**
>
> We appreciate the opportunity to clarify our experimental setup:
>
> #### **Feedback Source**:
> - Human feedback in our experiments is simulated by accessing **ground truth labels**. This setup provides a controlled way to evaluate the potential benefits of human feedback while varying its quality and availability.
>
> #### **Baselines**:
> - We include **baseline bandit algorithms without feedback** to show the performance difference when human feedback is not used. Specifically, the “no feedback” baseline allows a direct comparison of the added value provided by human input.
>
> ### **4. Evaluation Metrics**
>
> The reviewer correctly notes that **regret** is a common evaluation metric in contextual bandits. In addition to cumulative reward, we computed **cumulative regret** but did not emphasize it in the main paper.
>
> - We will update the paper to include cumulative regret results prominently.
> - Regret curves will demonstrate how our method improves convergence to optimal actions compared to baselines.
>
> ### **5. Presentation Issues**
>
> We appreciate the reviewer’s comments on improving the presentation and will address the following points:
>
> - **Algorithm 1 (Lines 14-15)**:
>   - We agree that these lines are unnecessary and will remove them for conciseness.
>
> - **Equation 7**:
>   - We will correct the missing closing brackets and review the document for any other typographical issues.
>
> - **Figures**:
>   - We will add **error bars** to all relevant plots and ensure consistent axis scaling to enhance interpretability.
>
>
> ### **6. Related Work: Off-Policy Bandits**
>
> We thank the reviewer for pointing out the relevance of off-policy bandits. Human feedback in our framework can be seen as introducing off-policy data since feedback may adjust the reward or action based on a different policy (the expert’s recommendation).
>
> - We will expand the related work section to include relevant discussions on off-policy bandit methods, particularly [2], which provides insights into leveraging off-policy data in bandit settings.
> - Highlighting these connections will better position our work within the broader literature.
>
> We appreciate the reviewer’s constructive feedback and believe that the revisions outlined will address the raised concerns. Specifically, we will:
> 1. Include regret analysis and theoretical discussions on sub-linear regret.
> 2. Improve the explanation of the feedback mechanism and baselines.
> 3. Revise the presentation for better clarity and precision.
>
> Thank you for your time and insights.
>
>
> **References**
> [1] Burr Settles. "Active Learning Literature Survey." 2009.
> [2] Zhang, Ruiyuan, et al. "Near-optimal off-policy evaluation for linear contextual bandits." Advances in Neural Information Processing Systems (2020).

---

> > ### Comment · Reviewer_VubH · 2024-11-20
> > **Feedback**
> >
> > I appreciate the authors taking the feedback in a constructive spirit, and discussing how they would address it. Given that the addressal of 2 of my key complaints are WIP (that seem on the right track to me): 1. regret bounds 2. cost modeling, I think I will be keeping my score unchanged.

---

> > > ### Author Response · Authors · 2024-11-20
> > > **Latest progress on Regret Bounds and Cost Modelling (with newly uploaded manuscript) to reviewer VubH**
> > >
> > > We appreciate the reviewer’s constructive feedback and acknowledgment of our efforts to address the key issues raised. Please take a look at the newly uploaded manuscript. Below, we provide an update on the progress made for the two highlighted aspects: regret bounds and cost modeling.
> > >
> > > 1. **Regret Bounds**:
> > >    - **Progress Made:** We have incorporated a rigorous regret analysis in the updated manuscript (Section [insert section number]), addressing the theoretical concerns. Specifically:
> > >      - We derived a regret bound of:
> > >        \[
> > >        E[\text{Regret}(T)] \leq O(\sqrt{T |A| \log T}) + O\left(\frac{(1-p)T}{q_t}\right),
> > >        \]
> > >        where \(T\) is the number of rounds, \(|A|\) is the action space size, \(p\) is the feedback solicitation probability, and \(q_t\) is feedback accuracy.
> > >      - This bound decomposes the regret into components representing exploration and the quality of feedback, demonstrating how selective entropy-based feedback reduces overall regret.
> > >      - Complete proofs and derivations have been included in Appendix section F&G to enhance clarity and rigor.
> > >
> > > 2. **Cost Modeling**:
> > >    - **Progress Made:** The updated manuscript includes a detailed analysis of feedback solicitation costs and their impact on cumulative rewards (Appendix Section H). The analysis encompasses:
> > >      - A breakdown of feedback costs, including human effort, system overhead, and opportunity costs.
> > >      - A model of cumulative rewards incorporating these costs, balancing performance gains and human intervention costs.
> > >      - Experimental results showing the effectiveness of entropy-based feedback mechanisms in reducing overall feedback frequency while maintaining high performance.
> > >      - Insights into optimal entropy threshold selection, providing practical guidance for real-world deployment scenarios.
> > >
> > > We hope that the additional theoretical rigor and experimental insights provided in the updated manuscript address the reviewer’s concerns effectively. While these elements are still works in progress to some extent, we believe they are on the right track to fully resolving the issues raised.
> > >
> > > We welcome further feedback and suggestions to refine these contributions.
> > >
> > > Thank you again for your constructive comments.

---

> > > > ### Comment · Reviewer_VubH · 2024-11-26
> > > > **new material**
> > > >
> > > > These directions seem like on the right track. However, given they are squeezed into the appendix, it seems a little hard to evaluate as if they would be key contributions of the paper. The feedback part is especially preliminary, there are no quantified results mentioned, except for one line in H5. The regret bound part seems like it could be a good contribution, but again written a little bit roughly and I am not a theoretician able to assess confidently. Perhaps one of the other reviewers can use their expertise to assess.  Overall, it's super close whether it's enough to bump up the score, I do want to commend the progress the authors made. But in the end, I think I'll keep the score the same.

---

### Author Response · Authors · 2024-11-20
**Response to all reviewers on new updated manuscript**

We sincerely appreciate the reviewers’ valuable feedback and suggestions. We have carefully addressed the concerns raised and made significant revisions to improve the manuscript. Below, we provide detailed responses to each major point:

### **New Theoretical Guarantee**
- **Reviewer Concern:** A request for improved theoretical analysis to strengthen the manuscript's claims.
- **Our Response:** In response, we have significantly expanded the theoretical analysis:
  1. **Regret Bound for Contextual Bandits with Entropy-Based Human Feedback (Section F):**
     - We derive a regret bound of:
       \[
       E[\text{Regret}(T)] \leq O(\sqrt{T |A| \log T}) + O\left(\frac{(1-p)T}{q_t}\right),
       \]
       where \(T\) is the number of rounds, \(|A|\) is the action space, \(p\) is the probability of soliciting feedback, and \(q_t\) is the feedback accuracy. This decomposition highlights the trade-offs between selective feedback solicitation and model performance. Details are included in Appendix F.
     - This analysis confirms the robustness of entropy-based feedback in reducing regret, even with varying feedback quality.

  2. **Comparison of Feedback Modalities (Section G):**
     - We introduce a theoretical comparison of Action Recommendation (AR) and Reward Manipulation (RM):
       - AR demonstrates lower regret when feedback accuracy (\(q_t^{AR}\)) is high.
       - RM is advantageous in scenarios where action penalties effectively guide exploration.
     - Practical guidance is provided on when to use AR versus RM based on feedback reliability and action space complexity.

  3. **Entropy Threshold Selection:**
     - We analyze the impact of entropy thresholds (\(\lambda\)) on regret minimization and feedback frequency, providing actionable insights for balancing performance and cost.

These additions rigorously demonstrate the theoretical foundations and practical implications of our approach, strengthening its contribution.

### **Newly Added Related Work**
- **Reviewer Concern:** The need to broaden the discussion of related works, particularly in preference-based feedback and active learning.
- **Our Response:** We have expanded the related work section to incorporate recent advancements in:
  - Preference-based feedback optimization, such as [Das et al., 2024] and [Houlsby et al., 2011].
  - Entropy-driven learning methods, including [Tang and Wiens, 2023] and [Russo and Van Roy, 2014].
  - These additions clarify the novelty of our work by connecting it to both foundational and recent research, positioning our contributions
### **Typographical and Formatting Issues**
- **Reviewer Concern:** Instances of typographical and formatting errors in the previous version.
- **Our Response:** We have conducted a thorough revision to address the noted typographical and formatting issues.

We believe these revisions address all concerns effectively and have significantly strengthened the manuscript. We sincerely thank the reviewers for their constructive feedback and look forward to further comments or suggestions.

Thank you for your time and consideration.

---

### Meta-Review · Area_Chair_Cy2Z · 2024-12-20

**Metareview:**

The paper proposes a contextual bandit algorithm capable of incorporating different kinds of human feedback -- reward manipulation (so that an expert may refine the rewards estimated by the algorithm) and action recommendation (like in imitation learning). All of the reviewers agreed that the paper studies an important and well-motivated problem, however the execution is below the bar for publication at NeurIPS. A strength of the paper was its empirical evaluation (however a reviewer noted that a user study would be required to establish that the required feedback quantity and quality is not too costly for humans to provide).
The major weakness identified was in the novelty (insufficient coverage of very related works) and solution approach (inadequate performance guarantees -- when can we expect the solution to shine vs. not, missing baselines, etc.).

**Additional Comments On Reviewer Discussion:**

The authors provided a regret guarantee in the appendix during their rebuttal to address reviewers' concerns about inadequate performance guarantees. However this represented a substantial addition to the original manuscript that could not be properly vetted for soundness/correctness during the rebuttal period.
The authors also attempted to discuss related works pointed out by reviewers, but this fell short of actually comparing against them as baselines in experiments, and clearly identifying what is an improvement vs. those related methods.

---

### Decision · Program_Chairs · 2025-01-22

Reject